# Integrative and quantitative view of the CtrA regulatory network in a stalked budding bacterium

Oliver Leicht[1©], Muriel C. F. van Teeseling[1©], Gaël Panis[2], Celine Reif[1], Heiko Wendt[1], Patrick H. Viollier[2], Martin Thanbichler[1,3,4]*

**1** Department of Biology, University of Marburg, Marburg, Germany, **2** Department of Microbiology and Molecular Medicine, Institute of Genetics and Genomics in Geneva (iGE3), Faculty of Medicine/CMU, University of Geneva, Geneve, Switzerland, **3** Center for Synthetic Microbiology (SYNMIKRO), Marburg, Germany, **4** Max Planck Institute for Terrestrial Microbiology, Marburg, Germany

© These authors contributed equally to this work.
* thanbichler@uni-marburg.de

**Data Availability Statement:** ChIP-seq and RNA-seq data have been deposited to the Gene Expression Omnibus (GEO) database under the accession code GSE134367. All other data

## Abstract

The *Alphaproteobacteria* show a remarkable diversity of cell cycle-dependent developmental patterns, which are governed by the conserved CtrA pathway. Its central component CtrA is a DNA-binding response regulator that is controlled by a complex two-component signaling network, mediating distinct transcriptional programs in the two offspring. The CtrA pathway has been studied intensively and was shown to consist of an upstream part that reads out the developmental state of the cell and a downstream part that integrates the upstream signals and mediates CtrA phosphorylation. However, the role of this circuitry in bacterial diversification remains incompletely understood. We have therefore investigated CtrA regulation in the morphologically complex stalked budding alphaproteobacterium *Hyphomonas neptunium*. Compared to relatives dividing by binary fission, *H. neptunium* shows distinct changes in the role and regulation of various pathway components. Most notably, the response regulator DivK, which normally links the upstream and downstream parts of the CtrA pathway, is dispensable, while downstream components such as the pseudokinase DivL, the histidine kinase CckA, the phosphotransferase ChpT and CtrA are essential. Moreover, CckA is compartmentalized to the nascent bud without forming distinct polar complexes and CtrA is not regulated at the level of protein abundance. We show that the downstream pathway controls critical functions such as replication initiation, cell division and motility. Quantification of the signal flow through different nodes of the regulatory cascade revealed that the CtrA pathway is a leaky pipeline and must involve thus-far unidentified factors. Collectively, the quantitative system-level analysis of CtrA regulation in *H. neptunium* points to a considerable evolutionary plasticity of cell cycle regulation in alphaproteobacteria and leads to hypotheses that may also hold in well-established model organisms such as *Caulobacter crescentus*.

supporting this study are provided in the Supplementary Material.

**Funding:** This study was funded by the German Research Foundation (DFG) (Project 192445154 - SFB 987; to M.T.), the Swiss National Science Foundation (grant 31003A_182576; to P.H.V) and the Max Planck Society (Max Planck Fellowship; to M.T.). M.C.F.v.T. was supported by a Long-term Postdoctoral Fellowship from the European Molecular Biology Organization (ALTF 1396-2015). The funders did not play any role in the design of the study, the collection and analysis of the data, the decision to publish, or the preparation of the manuscript.

**Competing interests:** The authors have declared that no competing interests exist.

## Author summary

Bacteria show a variety of morphologies and life cycles. This is especially true for members of the *Alphaproteobacteria*, a bacterial class of considerable ecological, medical, and biotechnological importance. The alphaproteobacterial cell cycle is regulated by a conserved regulatory pathway mediated by CtrA, a DNA-binding response regulator that acts as a transcriptional regulator and repressor of replication initiation. CtrA controls the expression of many genes with critical roles in cell growth, division, and differentiation. The contribution of changes in the CtrA regulatory network to the diversification of alphaproteobacterial species is still incompletely understood. Therefore, we comprehensively studied CtrA regulation in the stalked budding bacterium *Hyphomonas neptunium*, a morphologically complex species that multiplies by forming buds at the end of a stalk-like cellular extension. Our results show that this distinct mode of growth is accompanied by marked differences in the importance and subcellular localization of several CtrA pathway components. Moreover, quantitative analysis of the signal flow through the pathway indicates that its different nodes are less tightly connected than previously thought, suggesting the existence of so-far unidentified factors. Our results indicate a considerable plasticity of the CtrA regulatory network and reveal novel features that may also apply to other alphaproteobacterial species.

## Introduction

The ability to initiate processes such as DNA replication at the right moment in the cell cycle is crucial for the fitness of all cells. To ensure the correct timing of events, cells need to sense multiple input signals and integrate these signals in order to determine whether or not to start a certain process. A common strategy in both eukaryotes and prokaryotes is to use protein phosphorylation as a switching mechanism in this cellular information switchboard [1–3]. The integration of different signals often involves complex phosphorylation cascades, in which the reception of a signal leads to the phosphorylation of proteins, which then in turn phosphorylate downstream targets. These systems are called phosphorelays, or two component systems (TCSs) when the context is limited to a sensory histidine kinase and a downstream response regulator [4, 5]. Phosphorelays and TCSs play an important role in the regulation of cellular development of bacteria, controlling processes such as sporulation [6, 7] and cell cycle regulation [8, 9].

Many bacteria grow and divide without major changes in their overall morphology. However, there are also various species, including members of the *Alphaproteobacteria*, that undergo dramatic morphological changes when they proceed through their (asymmetric) cell cycle [10, 11]. In the alphaproteobacterial model organism *Caulobacter crescentus*, the two daughter cells have different shapes and fates: the stalked cell is sessile and can produce offspring, whereas the swarmer cell is motile and non-reproductive [9, 12]. This asymmetry is mainly dictated by the differential activation of the DNA-binding response regulator CtrA, whose phosphorylated form (CtrA~P) specifically accumulates in the swarmer but not in the stalked sibling [13, 14]. CtrA~P regulates, both positively and negatively, global gene expression by interacting with ~100 different promoter regions [15]. Apart from that, it binds to sites at the chromosomal origin of replication, thereby preventing the transition of cells into S phase [16]. In concert with other global regulators, such as the replication initiator/transcription factor DnaA [17–19] and the σ⁷⁰ cofactor GcrA [20, 21], CtrA thus ensures the ordered

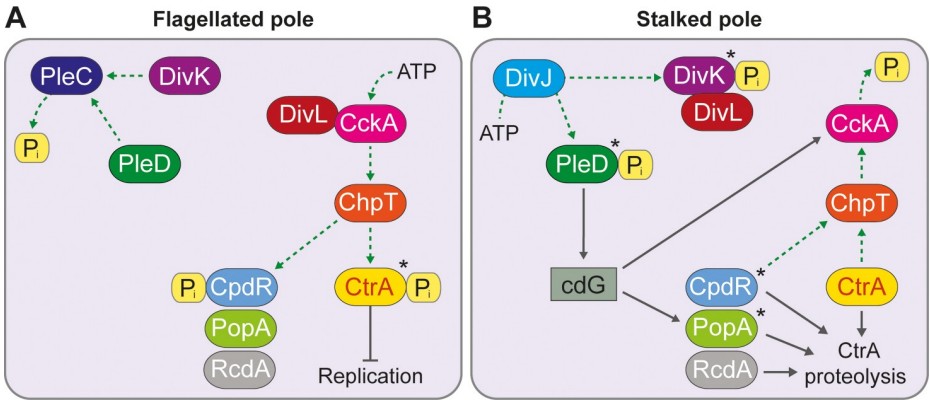

**Fig 1. Model of the cell-cycle dependent activation and stabilization of CtrA in *C. crescentus*.** **(A)** At the flagellated pole, PleC dephosphorylates PleD and DivK. DivL activates the kinase activity of CckA, which leads to the phosphorylation of CtrA via ChpT and, thus, to the activation of CtrA. ChpT also phosphorylates and thereby inactivates CpdR. **(B)** DivJ localizes to the stalked pole where it phosphorylates PleD and DivK. DivK~P sequesters DivL and thereby prevents DivL from activating the kinase activity of CckA. PleD~P synthesizes cyclic-di-GMP (cdG), which activates the phosphatase activity of CckA. As consequence, CckA dephosphorylates CtrA and CpdR. CpdR, together with PopA (activated by cdG made by PleD~P) and RcdA, stimulates the degradation of CtrA by ClpXP. The active forms of some of the proteins are marked with an asterisk (*).

progression of cells through their developmental program, licensing processes such as motility and cell division at the right moment in the cell cycle.

CtrA itself is regulated at the level of activity and abundance, and both of these parameters are affected by phosphorylation. The activation of CtrA proceeds via a phosphorelay including the histidine kinase CckA and the phosphotransferase ChpT [22, 23]. The different fates of the daughter cells result from the differential control of CckA activity at the two cell poles by the pseudokinase DivL and the single-domain response regulator DivK. At the flagellated pole, DivL interacts with CckA and stimulates its kinase activity, thereby mediating CtrA phosphorylation. At the stalked pole, by contrast, DivL is sequestered by DivK~P, which facilitates the transition of CckA to the phosphatase mode and the dephosphorylation of CtrA by reversion of the CckA-ChpT pathway [24, 25] (**Fig 1**). At the early predivisional stage, this cellular asymmetry is further enhanced by the preferential accumulation of CckA at the new cell pole [26, 27].

Phosphotransfer also plays an important role in the upstream pathway, which modulates the activity state of DivK. At the stalked pole, the localization factor SpmX [28, 29] recruits the histidine kinase DivJ [30], which in turn phosphorylates DivK and thereby enables the transition of CckA to the phosphatase mode [24, 25]. In parallel, DivJ phosphorylates the guanylate cyclase PleD, which then synthesizes the second messenger cyclic-di-GMP (c-di-GMP) [31] and, in this way, further stimulates the phosphatase activity of CckA [32]. At the flagellated pole, by contrast, the localization factor PodJ recruits the histidine kinase/phosphatase PleC [29, 33, 34], which dephosphorylates both DivK and PleD, thus triggering transition of CckA to the kinase state [24, 25]. Apart from that, PodJ is also involved in the positioning of DivK, DivL and CckA to the flagellated pole in predivisional cells [12].

In addition to its inactivation by dephosphorylation, *C. crescentus* CtrA is also subject to targeted proteolysis by the ClpXP protease complex [35, 36]. This process is activated by two convergent pathways that are both initiated by the histidine kinase DivJ. On the one hand, the reversal of the CckA-ChpT pathway induced by the phosphorylation of DivK not only dephosphorylates CtrA but also the adapter protein CpdR, which then primes ClpXP for CtrA degradation [37, 38] (**Fig 1**). On the other hand, c-di-GMP formed upon phosphorylation of PleD

binds and activates a second adapter protein, PopA, which then recruits CtrA to the primed ClpXP protease, triggering its degradation [39, 40].

Apart from *C. crescentus*, the *Alphaproteobacteria* include various other species with intriguing cell biological features [11]. The extensive body of knowledge on CtrA-dependent cell cycle regulation in *C. crescentus* has therefore sparked investigations into the conservation of this central pathway, with the aim to understand how its architecture evolved to adapt to different growth modes. These studies have shown that the CtrA pathway is essential in the polarly growing *Rhizobiales* [41–44], but dispensable in related species that divide by symmetric binary fission, such as *Rhodobacter capsulatus* and *Sphingomonas melonis* [45–48]. However, the CtrA pathway is still uninvestigated in the stalked budding alphaproteobacteria, a group of organisms that also includes the newly established model organism *Hyphomonas neptunium* [49, 50]. *H. neptunium* is closely related to *C. crescentus* [51] and shows a biphasic life cycle in which a motile swarmer cells sheds its flagellum, establishes a stalk and finally produces a daughter cell at the end of this stalk by expansion of the terminal stalk segment (stalk-terminal budding) [10, 52, 53] (**Fig 2**). It is currently unknown how the cell cycle regulatory circuitry has changed during evolution to give rise to this unusual mode of proliferation. Clarification of this issue may identify conserved features that have so far been neglected in well-studied model organisms such as *C. crescentus*, because they have become redundant in these species. Additionally, it may shed light on the plasticity of cell cycle regulatory networks and its role in the striking morphological diversification that alphaproteobacteria underwent during their adaptation to distinct environmental niches.

In this study, we show that the entire cell cycle regulatory pathway of *C. crescentus*, starting from the polarity-determining factors DivJ and PleC down to CtrA, is conserved in *H. neptunium*. Its downstream part, consisting of DivL, CckA, ChpT and CtrA, is essential and controls various critical functions, including DNA replication, cell division and flagellar synthesis. Strikingly, the upstream part of this pathway appears largely redundant, as DivK has no apparent effect on cell cycle progression and its interactors DivJ and PleC only make minor contributions. A quantification of the information flow through the entire CtrA pathway, using a novel approach that compares the regulons of the different nodes, suggests the existence of multiple so-far unidentified factors. These may include additional histidine kinases that are responsible for almost half of the input into the CtrA pathway as well as a response regulator

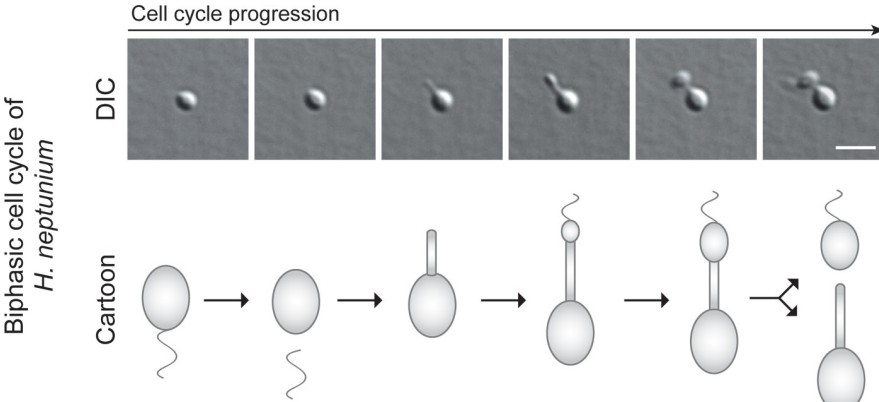

**Fig 2. Biphasic life cycle of *H. neptunium*.** Shown are a timelapse series (DIC) and a cartoon of a single *H. neptunium* cell that progresses through its cell cycle. The swarmer cell sheds its flagellum and differentiates into a stalked cell, which then produces a bud at the end of its stalk, thereby ultimately giving rise to a new swarmer cell. The stalked mother cell elongates its stalk and then immediately starts the next division cycle, whereas the swarmer cell first needs to differentiate into a stalked cell to proliferate. Scale bar: 3 μm.

that acts in parallel to DivK and that feeds signals from DivJ and PleC into a later part of the CtrA pathway. Localization studies reveal that CckA is distributed throughout the nascent daughter cell instead of forming a polar focus, which could have implications for the mechanism that CtrA uses to ensure asymmetry in the fate of the mother and daughter cell. Finally, we show that CtrA is not regulated at the level of protein abundance in *H. neptunium*. These results provide insight in the differential evolution of the cell cycle regulatory circuitry in *C. crescentus*, which divides by asymmetric binary fission, and *H. neptunium*, which divides by stalk-terminal budding. Furthermore, they set the stage for the identification of new factors that play a major role the regulation of the *H. neptunium* cell cycle and may also contribute to cell cycle progression and cellular differentiation in other alphaproteobacteria.

## Results

### The upper part of the CtrA pathway is functionally redundant, while the lower part is essential

The *H. neptunium* genome encodes homologs of all proteins directly involved in the CtrA pathway of *C. crescentus* (Table 1). DivJ, PleC, DivK, DivL and CckA from *H. neptunium* could (partially) complement temperature-sensitive mutations in the corresponding factors in *C. crescentus* (S1A Fig), verifying that they are indeed *bona fide* homologs. Induction of CtrA from *H. neptunium* in either wild-type or *ctrA*ts strains of *C. crescentus* led to cell elongation and eventually to lysis, even at the permissive temperature (S1B–S1D Fig), suggesting that the CtrA homologs of these two organisms have functionally diverged, or respond differently to regulation. To test the importance of DivJ, PleC, DivK, PleD, CckA, ChpT and CtrA in *H. neptunium*, we set out to generate in-frame deletions in the respective genes. Notably, mutants were obtained for all genes belonging to the upstream part of the pathway (*divJ*, *pleC*, *divK* and *pleD*). An analysis of these strains for potential cell cycle-related phenotypes showed that most

**Table 1. *H. neptunium* homologs of cell cycle-related genes from *C. crescentus*.** The table summarizes the query coverage, e-value and degree of identity for the best hits in *H. neptunium* obtained in BLAST searches with the respective proteins from *C. crescentus* as a query. The e-values of the respective second-best hits are given for comparison. The information in the column 'Best reciprocal hit?' indicates whether the indicated *C. crescentus* protein is the best hit in a BLAST search performed with the respective *H. neptunium* homolog as a query. NA indicates that no further hits were detected.

| Gene | Locus tag *C. crescentus* | Locus tag *H. neptunium* | Query coverage (%) homolog | e-Value homolog | Identity (%) homolog | Best reciprocal hit? | e-Value alternative hit |
|------|-----|-----|-----|-----|-----|-----|-----|
| *cckA* | CCNA_01132 | HNE_0507 | 74 | $5 \times 10^{-143}$ | 47 | yes | $1 \times 10^{-56}$ |
| *chpT* | CCNA_03584 | HNE_0638 | 82 | $1 \times 10^{-23}$ | 26 | yes | NA |
| *clpP* | CCNA_02041 | HNE_2087 | 98 | $7 \times 10^{-113}$ | 72 | yes | $2 \times 10^{-42}$ |
| *clpX* | CCNA_02039 | HNE_2086 | 98 | 0.0 | 82 | yes | $1 \times 10^{-24}$ |
| *cpdR* | CCNA_00781 | HNE_0229 | 100 | $1 \times 10^{-50}$ | 66 | yes | $2 \times 10^{-19}$ |
| *ctrA* | CCNA_03130 | HNE_0944 | 96 | $6 \times 10^{-132}$ | 76 | yes | $1 \times 10^{-41}$ |
| *divJ* | CCNA_01116 | HNE_0746* | 40 | $2 \times 10^{-43}$ | 43 | no** | $1 \times 10^{-60}$ |
| *divK* | CCNA_02547 | HNE_2285 | 99 | $2 \times 10^{-57}$ | 60 | yes | $1 \times 10^{-19}$ |
| *divL* | CCNA_03598 | HNE_0399 | 84 | $2 \times 10^{-162}$ | 41 | yes | $2 \times 10^{-26}$ |
| *pleC* | CCNA_02567 | HNE_2910 | 68 | $5 \times 10^{-125}$ | 42 | yes | $2 \times 10^{-40}$ |
| *pleD* | CCNA_02546 | HNE_2284 | 100 | $2 \times 10^{-143}$ | 50 | yes | $5 \times 10^{-40}$ |
| *podJ* | CCNA_02125 | HNE_0666 | 73 | $3 \times 10^{-36}$ | 38 | yes | NA |
| *popA* | CCNA_01918 | HNE_1984 | 80 | $8 \times 10^{-14}$ | 25 | yes | 0.49 |
| *rcdA* | CCNA_03404 | HNE_0013 | 88 | $1 \times 10^{-39}$ | 48 | yes | 7.6 |
| *spmX* | CCNA_02255 | HNE_1271 | 52 | $8 \times 10^{-45}$ | 44 | yes | 1.9 |

\* The homolog of *divJ* was predicted based on gene synteny.

\*\* HNE_0746 is not the best hit for CCNA_01116, but CCNA_01116 is the best hit for HNE_0746.

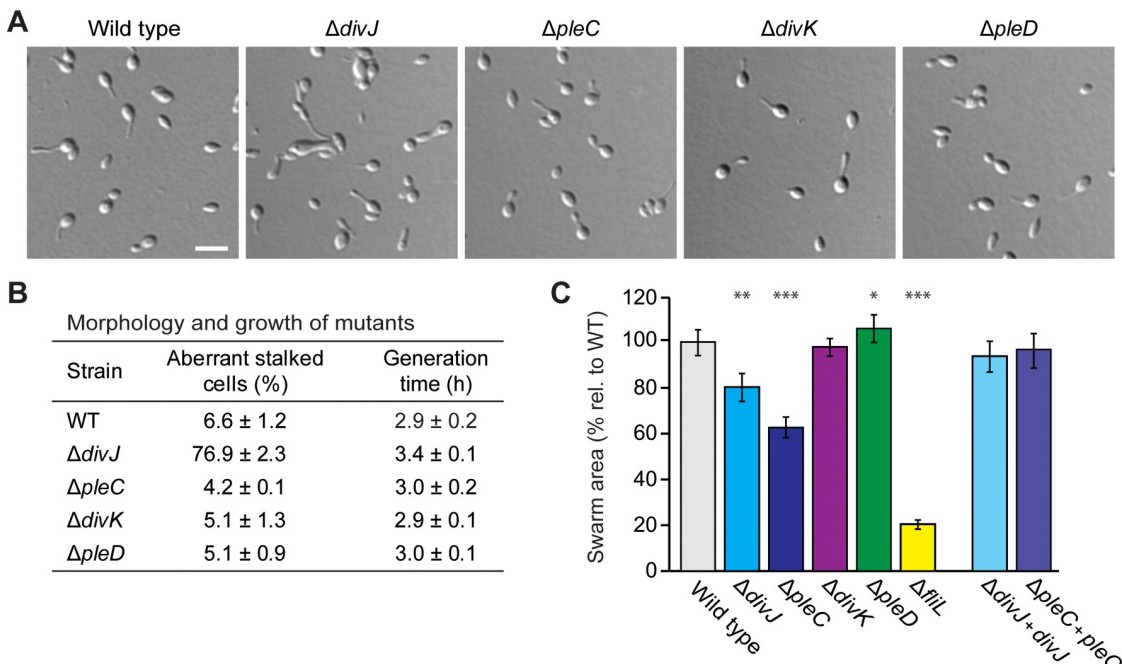

**Fig 3. DivJ, PleC, DivK and PleD have minor effects on *H. neptunium* morphology and motility.** **(A)** DIC images of the *H. neptunium* wild type and mutants lacking DivJ (OL94), PleC (OL20), DivK (OL95) and PleD (OL21). The scaling of all images is identical. Scale bar: 3 μm. **(B)** Quantification of the morphological defects and the generation times of the strains described in (A). The number of cells analyzed to determine the percentage of aberrant stalk cells is 454 (wt), 438 (ΔdivJ), 430 (ΔpleC), 437 (ΔdivK), and 439 (ΔpleD). The generation times given represent the mean values from at least four replicate growth curves. **(C)** Motility of *H. neptunium* cells lacking DivJ, PleC, DivK or PleD. A mutant lacking the flagellar protein FliL (HW2) was used as a negative control. Strains complemented for DivJ (OL123) and PleC (OL23) were analyzed to verify that the motility defects of the mutant strains are caused by the lack of DivJ or PleC, respectively. To quantify motility, the indicated strains were spotted on soft agar and incubated for 6 days prior to imaging and quantification of the area of growth. In case of the DivJ and PleC complementation strains, inducer (300 μM $CuSO_4$) was added to the medium. The swarm areas given are normalized to the value obained for the wild-type strain and represent the mean of at least fourteen replicates. Asterisks indicate statistically significant differences between the wild type and the respective mutant strain (one-sided Mann Whitney U-test): * p-value <0.05, ** p-value <0.01 and *** p-value <0.001.

of them exhibited wild-type morphologies and grow rates (**Fig 3A and 3B**). The Δ*divJ* mutant, by contrast, grew slightly more slowly, and the majority of the stalked cells displayed an aberrant morphology, as reflected by elongated stalks and/or swollen cell bodies (**Fig 3A and 3B**). Apart from that, the Δ*divJ* and Δ*pleC* strains had clear motility defects, whereas the Δ*divK* and Δ*pleD* mutants still showed wild-type swimming behavior (**Fig 3C**). The motility defects of the Δ*divJ* and Δ*pleC* strains and the growth and morphology defects of the Δ*divJ* strain could be complemented by expression of the respective genes from an inducible promoter (**Fig 3C and S2 Fig**). Collectively, these results show that the DivJ/PleC-DivK module only has a minor effect on cell cycle regulation in *H. neptunium*. This finding is in stark contrast to the situation in *C. crescentus*, where DivK is essential [54] and DivJ [29, 55] as well as PleC [29, 56] play prominent roles in the control of cellular asymmetry and development.

Unlike for the upstream part of the CtrA pathway, it was impossible to obtain *H. neptunium* deletion mutants lacking components of the downstream pathway, including DivL, CckA, ChpT and CtrA. To verify the essentiality of these proteins, we aimed to express their genes under the control of an inducible promoter and then study the effects of their depletion. This approach failed in the case of CtrA, possibly due to the high expression level of *ctrA* and the limited strength of the inducible promoters established for *H. neptunium* [49]. However, we did succeed in constructing conditional *divL*, *cckA*, and *chpT* mutants. Under restrictive conditions, all three strains showed a highly aberrant, heterogeneous morphology, with a high

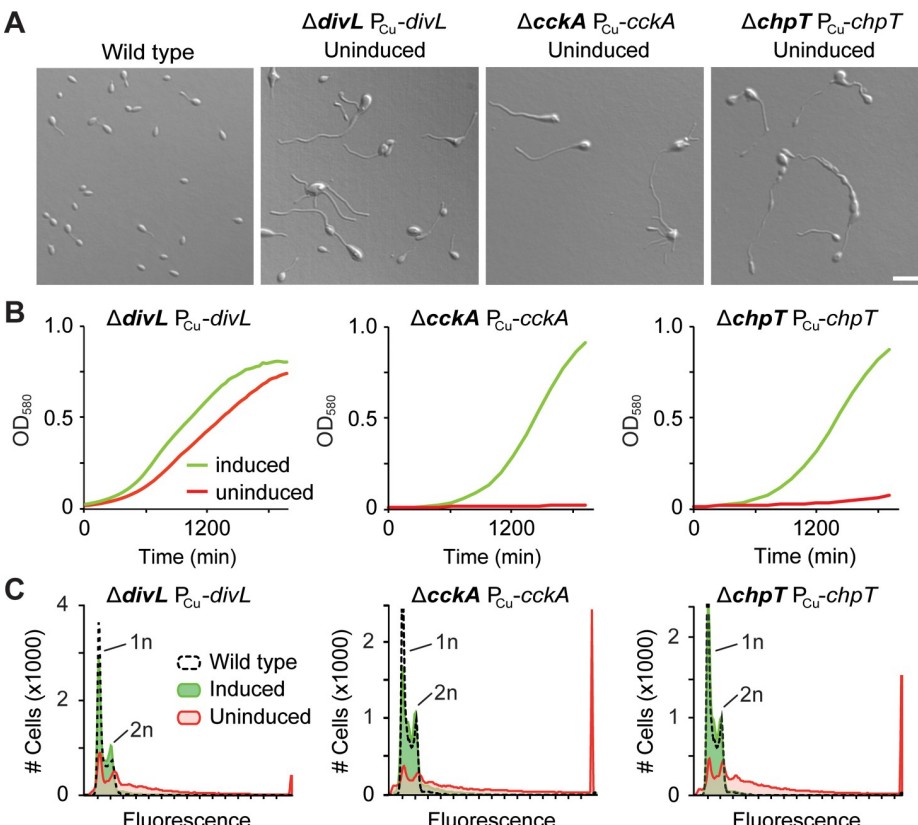

**Fig 4. The essential proteins DivL, CckA and ChpT affect cell morphology and chromosome content. (A)** DIC images of the *H. neptunium* wild type and conditional mutants depleted of DivL (OL177), CckA (OL161) and ChpT (OL152). Protein depletion was achieved by cultivation of the strains for at least 24 h in the absence of inducer (300 μM $CuSO_4$). The scaling of all images is identical. Scale bar: 5 μm. **(B)** Growth of conditional *divL* (OL177), *cckA* (OL161) and *chpT* (OL152) mutants under inducing (300 μM $CuSO_4$) and non-inducing conditions. The curves shown represent the mean of three independent experiments. **(C)** Flow cytometric analysis showing the DNA content of the indicated conditional mutants under inducing and non-inducing conditions (n = 30,000 cells per conditions). Data obtained for the wild-type strain are shown for comparison. The fluorescence intensities corresponding to one (1n) or two (2n) chromosome equivalents are indicated.

incidence of severely enlarged cell bodies and elongated and/or ectopic stalks (**Fig 4A**). At the same time, they showed a considerable reduction in the growth rate, although the effect was less pronounced for DivL-depleted cells, possibly due to leaky expression of the *divL* gene (**Fig 4B**). Flow cytometric analysis revealed that these defects correlated with a considerable increase in the DNA content, with many cells accumulating far more than two chromosome equivalents (**Fig 4C**). Interestingly, despite their aberrant shapes, the mutant cells were still able to segregate DNA to the nascent daughter cell compartments (**S3 Fig**), suggesting that their elevated DNA content is caused by overreplication of the chromosome in the absence of cell division. Collectively, these results demonstrate that the downstream part of the CtrA pathway, including DivL, CckA, ChpT as well as CtrA, has an essential role in *H. neptunium* cell cycle regulation, whereas the upstream part is, in large part, dispensable.

## DivJ and PleC mark opposite poles, whereas CckA dynamically localizes to entire cell compartments

To further study the functional conservation of the CtrA pathway in *H. neptunium*, we decided to determine the localization patterns of DivJ, PleC and CckA. To this end, the corresponding

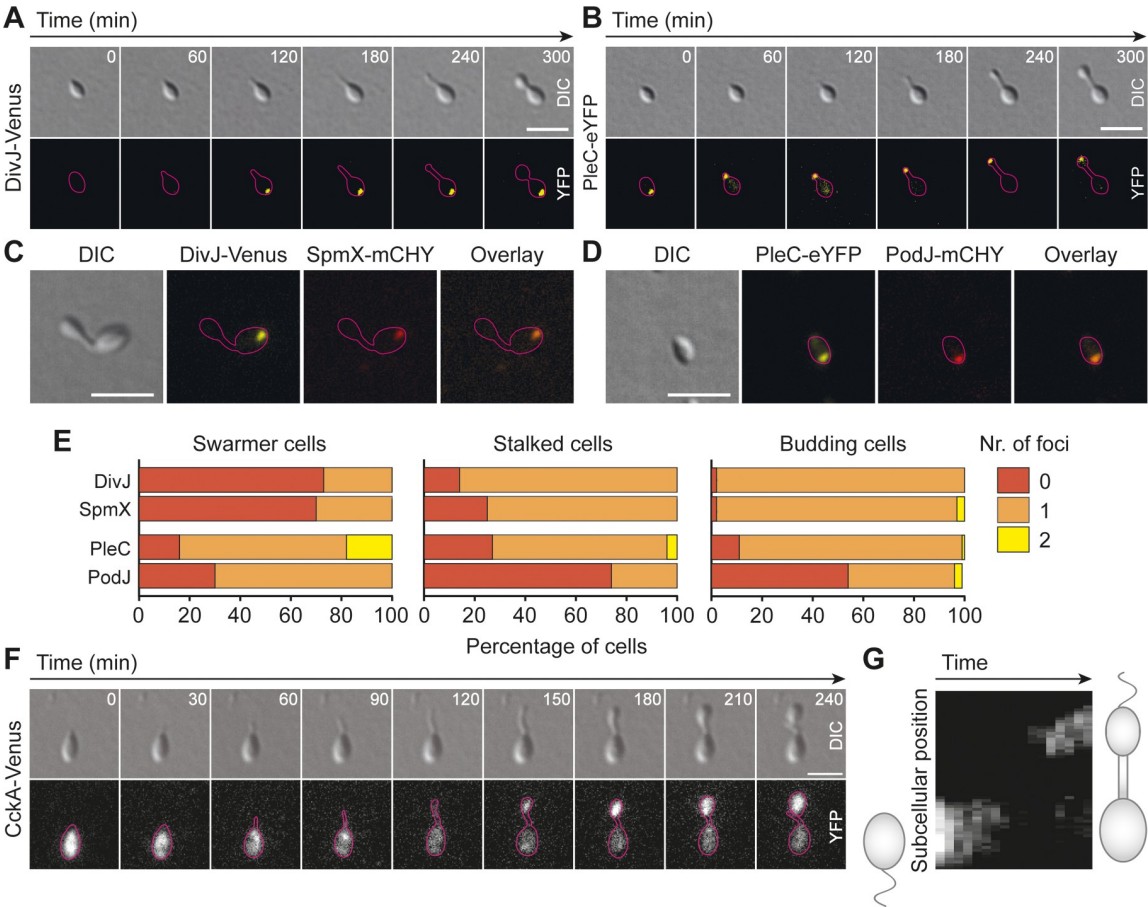

**Fig 5. DivJ and PleC mark opposite poles and CckA accumulates in the nascent bud. (A)** and **(B)** Timelapse experiments following *H. neptunium* cells producing (A) DivJ-Venus (OL146) or (B) PleC-eYFP (OL151) from their respective native promotors. **(C)** Colocalization of DivJ-Venus (expressed from its native promotor) and SpmX-mCherry (induced by addition of 300 μM CuSO₄) (OL32). **(D)** Colocalization of PleC-eYFP (expressed from its native promotor) and PodJ-mCherry (induced by addition of 300 μM CuSO₄) (OL33). **(E)** Quantification of the number of fluorescent foci in cells imaged as described in panels (C) and (D). For each strain, 100 cells were analyzed per developmental stage. Note that the intensity of the PodJ-mCherry signal was close to the detection limit. Therefore, the counts given for this fusion protein may be an underestimation, explaining the discrepancy in the number of PodJ-mCherry and PleC-eYFP foci. **(F)** Timelapse experiment following a representative *H. neptunium* cell expressing CckA-Venus from its native promotor (OL2). **(G)** Kymograph of the fluorescence signal in the cell depicted in (F). Shown is the maximum signal on a line connecting the two poles and spanning the entire width of the cell. Scale bars: 2 μm.

endogenous genes were replaced with alleles encoding fluorescent protein fusions. Time-lapse analysis revealed that DivJ is absent in most swarmer cells but then localizes to the old pole of early stalked cells, retaining this position until the end of the division cycle (**Fig 5A and 5E**). PleC, by contrast, is initially positioned at the old pole of the swarmer cell and then relocates to the tip of the growing stalk, so that it finally ends up at flagellated pole of the bud compartment (**Fig 5B and 5E**). Similar to their *C. crescentus* homologs, the two proteins thus mark opposite ends of the predivisional cell. Further analyses showed that DivJ and PleC frequently colocalize with the *H. neptunium* SpmX and PodJ homologs, respectively (**Fig 5C–5E**), and depend on their corresponding partner protein for proper localization (**S4 Fig**). Interestingly, in the absence of PodJ, most cells not only lacked a PleC focus but also became misshapen once they entered the stalked stage (**S4 Fig** and **S5 Fig**). As this effect was not observed for a Δ*pleC* mutant, PodJ may have important additional functions that go beyond its interaction with PleC. Unlike in the case of DivJ and PleC, the localization pattern of CckA clearly differed

between *H. neptunium* and *C. crescentus*. The *H. neptunium* homolog, a membrane protein containing two predicted transmembrane domains (like its homolog in *C. crescentus*), showed an even distribution in swarmer cells, assumed an irregular, patchy pattern in stalked cells, and finally accumulated in the nascent bud without forming a noticeable polar focus (**Fig 5F and 5G**). Importantly, cells producing fluorescently tagged CckA do not show any morphological or growth defect (**Fig 5** and **S6 Fig**). Thus, CckA appears to exert its function without accumulating at the pole in *H. neptunium*. Together, our results suggest that DivJ and PleC are involved in establishing cell polarity in *H. neptunium*, whereas CckA aids in the asymmetric activation of CtrA.

## The connectivity of the components constituting the CtrA pathway is conserved in *H. neptunium*

In *C. crescentus*, the localization of DivJ and PleC to opposite cell poles leads to the differential phosphorylation of their common target DivK and, thus, to the asymmetric activation of DivL and the downstream CckA-ChpT-CtrA phosphorelay in the two daughter cells [22, 23–25, 29, 30]. To clarify whether these interactions are conserved in *H. neptunium*, we aimed to analyze the connectivity of the different proteins using *in vivo* and *in vitro* approaches. In doing so, we first focused on the upstream part of the CtrA pathway (**Fig 6A**). Bacterial two-hybrid analysis confirmed that both DivJ and PleC interact with DivK and that DivK interacts with DivL (**Fig 6B**). Furthermore, PleC and DivL showed a strong self-interaction, as has been suggested for PleC and shown for DivL in *C. crescentus* [57, 58]. Since both DivJ and PleC are also able to phosphorylate DivK *in vitro* (**Fig 6C–6E**), we conclude that the protein-protein interactions within the upstream part of the CtrA pathway are conserved. Next, we aimed to confirm the functionality of the CckA-ChpT-CtrA phosphorelay (**Fig 7A**). Since it was not possible to achieve autophosphorylation of *H. neptunium* CckA *in vitro*, we resorted to its functionally equivalent *C. crescentus* homolog (CckA$_{CC}$) (see **S1A Fig**) as a phosphoryl donor. Our *in vitro* assays show that ChpT can be phosphorylated by CckA$_{CC}$ and subsequently transfer a phosphoryl group to CtrA (**Fig 7B and 7C**). Moreover, they confirm that the response regulator domain of *H. neptunium* CckA can phosphorylate ChpT after it has received a phosphoryl group from the autophosphorylated kinase domain of CckA$_{CC}$ (**Fig 7D and 7E** and **S7 Fig**). These results suggest that the signaling cascade involved in the activation of CtrA is conserved in *H. neptunium*, although *H. neptunium* CckA may need a specific trigger to switch to its kinase state.

## Global analyses indicate a central role of CtrA in the regulation of cell cycle-related processes

Having shown the conservation of the pathway mediating CtrA activation, we set out to identify the regulatory targets of CtrA in *H. neptunium*. For this purpose, we depleted cells of CckA or ChpT to inhibit CtrA phosphorylation (and induce its partial depletion; see **S8A Fig** and **S8B Fig**) and then performed global transcriptomic analyses to identify genes that were differentially regulated in these conditions. The expression profiles obtained for CckA- and ChpT-depleted cells were highly similar (**Fig 8A**), verifying that both proteins act in the same pathway. In total, 381 genes were significantly (at least 2.5-fold; p<0.05) up- or down-regulated in the absence of CckA and/or ChpT (**Fig 8A**, **S9A Fig** and **S1 Data**). We define these 381 genes as the global CtrA regulon, although it cannot be excluded that some of them are controlled by other regulators than CtrA. To pinpoint the genes that are directly regulated by CtrA, we determined the chromosomal binding sites of CtrA by chromatin immunoprecipitation and subsequent high-throughput sequencing of isolated DNA fragments (ChIP-seq) (**Fig 8B** and

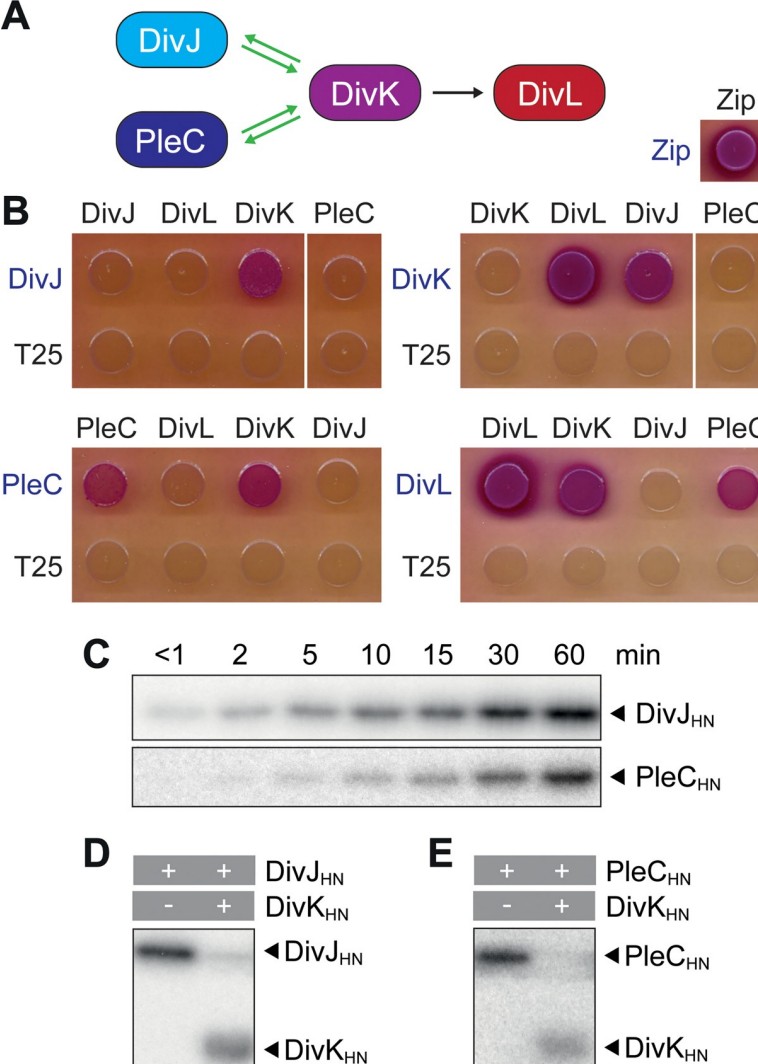

**Fig 6. The connectivity within the DivJ/PleC-DivK module is conserved in *H. neptunium*.** (A) Scheme of the interactions within the DivJ/PleC-DivK-DivL module, as demonstrated in this study. Green arrows indicate phosphotransfer. The black arrow indicates a regulatory protein-protein interaction. (B) Bacterial two-hybrid assay testing for interactions between the proteins in the DivJ/PleC-DivK-DivL. The indicated proteins were fused to the T18 (black) or T25 (blue) subunit of adenylate cyclase from *Bordetella pertussis*, respectively. The fusions were produced in the reporter strain *E. coli* BTH101, and their interaction was analyzed by spotting of the cells on MacConkey agar. Interactions are indicated by a purple coloration of the colonies. The GCN4 leucin-zipper-region from yeast (Zip) was used as positive control, and empty vectors encoding only the T25 subunit as negative controls. (C) Autophosphorylation of $DivJ_{HN}$ and $PleC_{HN}$ upon incubation with $[\gamma^{32}P]$-ATP. Samples were taken at the indicated time points and analyzed by SDS-PAGE. Radioactivity was detected by phosphor imaging. (D) and (E) *In vitro* phosphotransfer from (D) $DivJ_{HN}$ and (E) $PleC_{HN}$ to $DivK_{HN}$. After autophosphorylation of $DivJ_{HN}$ for 30 min and of $PleC_{HN}$ for 45 min, the indicated proteins (marked with pluses) were incubated for 90 sec. After addition of SDS sample buffer to stop the reactions, the mixtures were separated by SDS-PAGE, and radioactivity was detected by phosphor imaging.

S2–S5 Data). Out of the 222 sites identified in this analysis, 211 were located in intergenic regions, potentially affecting the expression of 285 genes (S6 Data). A comparison of the sequences bound by CtrA identified the consensus motif $TTAA-N_7-TTAAC$ (Fig 8C). Very similar motifs have been reported for CtrA homologs from other species [14, 15, 41, 42, 46, 59–61], which confirms the specificity of the ChIP-seq approach. Based on the ChIP-seq data, we then selected genes that carried a CtrA binding site in their coding or promoter region and

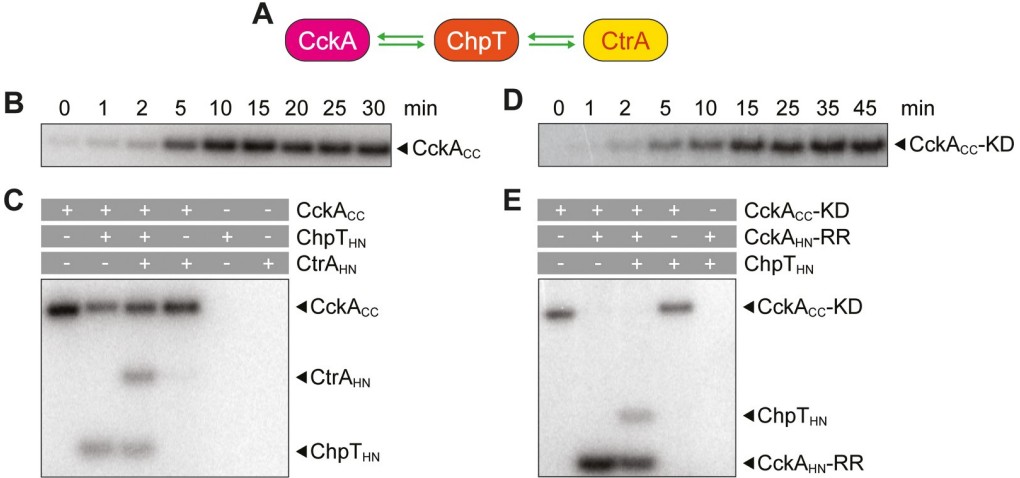

**Fig 7. The connectivity within the CckA-ChpT-CckA phosphorelay is conserved in *H. neptunium*. (A)** Scheme of the CckA-ChpT-CtrA phosphorelay. Green arrows indicate phosphotransfer. **(B)** Autophosphorylation of $CckA_{CC}$ upon incubation with $[\gamma^{32}P]$-ATP. Samples were taken at the indicated time points after the start of the reaction and analyzed by SDS-PAGE prior to detection of radioactivity by phosphor imaging. **(C)** *In vitro* phosphotransfer from $ChpT_{HN}$ to $CtrA_{HN}$ with $CckA_{CC}$ as a phosphoryl donor. After autophosphorylation of $CckA_{CC}$ for 30 min, the indicated proteins (marked with pluses) were mixed and incubated for 5 min. After termination of the reactions by addition of SDS sample buffer, the mixtures were separated by SDS-PAGE and radioactivity was detected by phosphor imaging. **(D)** Autophosphorylation of CckA-$KD_{CC}$ upon incubation with $[\gamma^{32}P]$-ATP. Samples were taken at the indicated time points and analyzed by SDS-PAGE prior to detection of radioactivity by phosphor imaging. **(E)** *In vitro* phosphotransfer from CckA-$RR_{HN}$ to $ChpT_{HN}$ with CckA-$KD_{CC}$ as a phosphoryl donor. After autophosphorylation of CckA-$KD_{CC}$ for 45 min, the indicated proteins (marked with pluses) were mixed and incubated for 5 min. The reaction was terminated by addition of SDS sample buffer, and the mixtures were separated by SDS-PAGE prior to detection of radioactivity by phosphor imaging.

showed differential regulation upon CckA/ChpT depletion (55 genes) or genes that were part of a putative operon that fulfilled these criteria (39 genes). The resulting 94 genes were defined as the direct CtrA regulon (**Fig 8D and 8E**, **S9C Fig** and **S7 Data**). Notably, several genes that are tightly bound by CtrA (**S6 Data**) are not part of the CtrA regulon. It is possible that the high-affinity binding sites in their promoters recruit non-phosphorylated CtrA or residual CtrA~P that is left after depletion of CckA or ChpT, leading to a lack of significant changes in expression under the experimental conditions used. Alternatively, since we analyzed mixed cultures, some CtrA-regulated genes may not be included in the global regulon because their mean expression levels (averaged over the cell cycle) were not significantly different from the levels obtained after CckA or ChpT depletion. Notably, the finding that many CtrA-bound genes are not differentially expressed upon inactivation of CtrA is not unique to *H. neptunium*, as a similar phenomenon has been previously observed for *C. crescentus* [15].

A functional categorization of the genes bound and/or regulated by CtrA revealed that many of them have no predicted function (37% of the direct regulon) or a putative role in 'cellular processes and signaling' (43% of the direct regulon) (**S9 Fig**). Specific functions that are extensively regulated by CtrA are flagellar motility (15 genes; 16% of the direct regulon) and pili-mediated adhesion (14 genes; 15% of the direct regulon) (**S7 Data** and **S8 Data**). Consistent with this result, a large majority of the genes and operons in the motility island of *H. neptunium* are preceded by a CtrA binding site (**Fig 9**). As in *C. crescentus* [62], pili genes are generally up-regulated by CtrA in *H. neptunium*. For flagellar genes, by contrast, the mode of regulation varies, suggesting a more complex regulatory mechanism. CtrA also appears to affect morphogenesis and cell division by directly controlling the expression of *mreB* and the cell division genes *ftsAKQZ* and *ftsB* (HNE_1978), respectively (**Fig 8E**). Moreover, it may

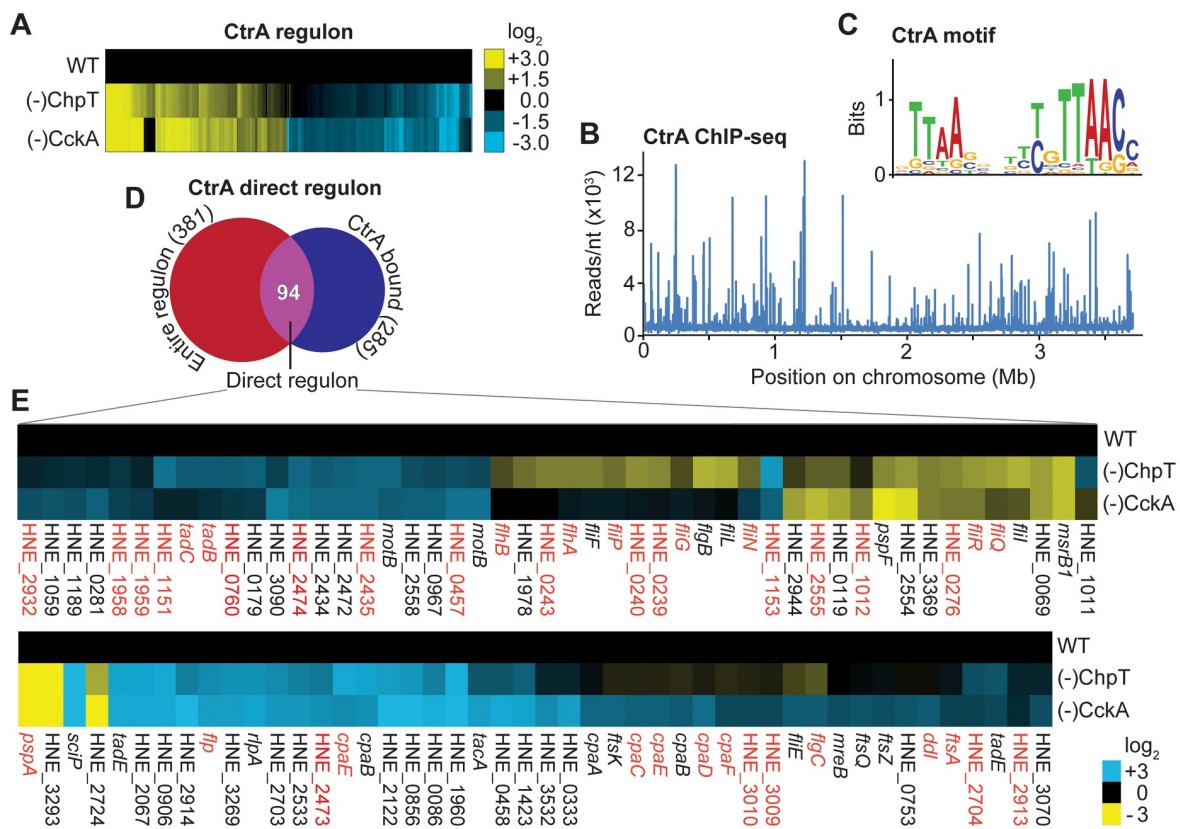

**Fig 8. The direct CtrA regulon contains 94 genes. (A)** Transcriptional profiles of cells depleted of CckA and ChpT for 24 h. Shown are the log$_2$-fold changes in the expression levels of all 381 genes that were differentially regulated upon disruption of the CckA-ChpT-CtrA phosphorelay (as compared to the wild-type strain). **(B)** Distribution of CtrA binding sites on the *H. neptunium* chromosome as identified by ChIP-seq analysis. The graph reveals a total of 222 CtrA binding sites, 211 of which are in intergenic regions. **(C)** CtrA consensus binding motif obtained by a comparison of the 80 chromosomal regions that were most enriched in the ChIP-seq data shown in (B). **(D)** Venn diagram showing the number of genes (single or in an operon) bound by CtrA (in blue) and the number of genes present in the entire CtrA regulon (in red). The intersection of these two gene sets defines the direct CtrA regulon and comprises 94 genes. **(E)** Direct CtrA regulon. Shown are the 94 genes contained in the direct CtrA regulon and the log$_2$-fold changes in their expression compared to the wild type after depletion (24 h) of ChpT or CckA. Only genes with an RPKM value of >25, a p-value of <0.05 and a log$_2$-fold difference of >2 were taken in account.

influence cell cycle progression by modulating the levels of the universal second messenger c-di-GMP, as it regulates eight genes that are predicted to be involved in c-di-GMP synthesis or degradation in both a direct (HNE_0906, HNE_1423, HNE_1960, HNE_2067, HNE_2435, HNE_2558) and indirect (HNE_0279, HNE_3504) fashion (**S7 Data** and **S8 Data**). Interestingly, as in *C. crescentus* [21, 63], CtrA also interacts with the promoter region of its own gene (which additionally contains a binding site for the global regulator GcrA), suggesting the existence of an autoregulatory feedback loop (**Fig 9B**). Moreover, it controls the gene for the CtrA co-regulator SciP, which in *C. crescentus* represses the transcription of *ctrA* and other CtrA targets [64, 65]. CtrA may thus also regulate its own transcription in an indirect fashion (**S7 Data** and **S8 Data**). However, while affecting its accumulation at the transcriptional level, CtrA does not regulate genes for factors known to mediate CtrA proteolysis in *C. crescentus*, such as CpdR, RcdA, PopA and ClpXP (**S7 Data**). Notably, CtrA does not only regulate a functionally diverse set of genes but it also binds to a site next to the chromosomal origin of replication (as identified in [66]) (**Fig 9C**), suggesting that it could contribute to the control of DNA replication.

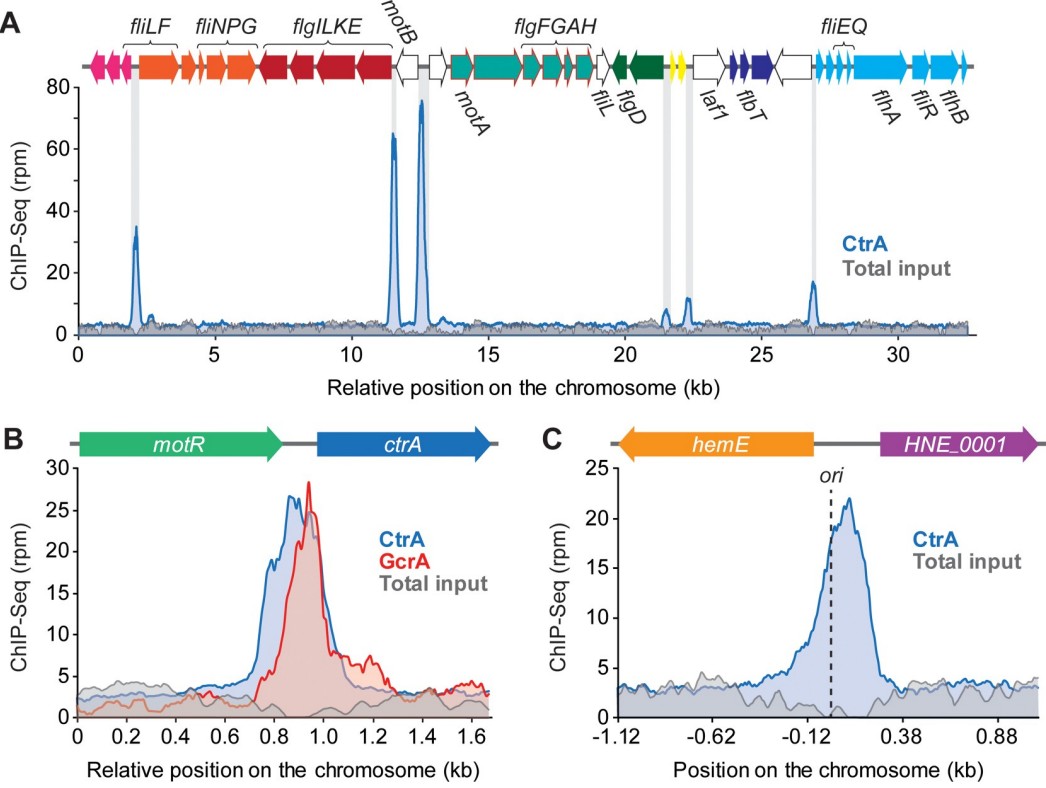

**Fig 9. CtrA binds near the replication origin, in its own promotor region and in the motility island. (A)** Distribution of CtrA binding sites in the motility island of *H. neptunium* (HNE_0239-HNE_0276). The graph shows the number of sequenced fragments per nucleotide in rpm (reads per million) for a ChIP-seq experiment performed with CtrA as a bait (blue). Sequencing results obtained for total input DNA (in grey) are presented as a negative control. The scheme on top provides an overview of the motility island. Genes predicted to be in the same operon are shown in the same color, orphan genes are indicated by white arrows. Genes or operons not preceded by a CtrA binding site are indicated by a red frame. **(B)** CtrA binding in the promoter region of the *ctrA_{HN}* gene. Shown are the normalized number of sequenced fragments (rpm) obtained in ChIP-seq experiments with CtrA (blue) and GcrA (red). Sequencing results obtained for total input DNA (grey) are shown as a negative control. **(C)** CtrA binding at the chromosomal origin of replication (*ori*). Colors as in (A).

## The DivJ/PleC-DivK module only has a minor effect on the activity of the CckA-ChpT-CtrA phosphorelay

Detailed knowledge of the CtrA targets does not only reveal the regulatory potential of this central cell cycle regulator in *H. neptunium*, but it also facilitates a quantitative assessment of the signal flow within the CtrA pathway. In particular, it offers the possibility to clarify the contribution of the non-essential DivJ/PleC-DivK module to the activity of the essential CckA-ChpT-CtrA phosphorelay. To this end, we determined the transcriptional profiles of the ΔdivJ, ΔpleC and ΔdivK mutants and compared them to those of the CckA and ChpT depletion strains (**Fig 10A and S9 Data**). In general, the three mutants showed considerably smaller changes in global gene expression than the two depletion strains, supporting the notion that the upstream part of the CtrA pathway has a smaller effect on the activity of CtrA than its downstream part. To enable a meaningful comparison of the data, we applied less stringent criteria to define the regulons of DivJ, PleC and DivK, using a p-value of <0.25 and a $\log_2$-fold difference of >0.5 as thresholds (instead of <0.05 and >1.3, respectively, as used for CckA and ChpT). These lower thresholds could potentially lead to the inclusion of some false positives in the respective regulons. Nevertheless, the DivJ, PleC and DivK regulons identified in this

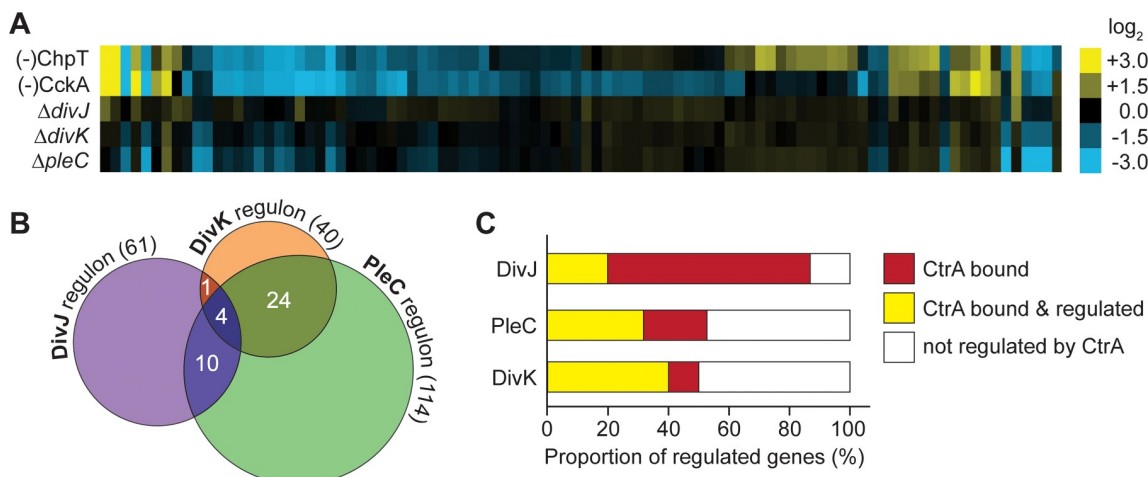

**Fig 10. DivJ, PleC and DivK only have a minor effect on the CckA-ChpT-CtrA pathway. (A)** Comparison of the transcriptional changes observed for genes in the direct CtrA regulon in the indicated depletion (-) or deletion (Δ) backgrounds. Shown are the log$_2$-fold changes in transcript levels in the mutant strains as compared to the wild type. **(B)** Venn diagram depicting the overlap between the regulons of DivJ, PleC and DivK. The number of genes in the respective regulons is given in parentheses. Only genes with an RPKM value of >25, a p-value of <0.25 and a log$_2$-fold difference of >0.5 were taken into account. **(C)** Proportion of genes in the regulons of DivJ, PleC and DivK that are regulated by CtrA. Genes present in the direct CtrA regulon are shown in yellow. Genes that are not included in the direct CtrA regulon but have a CtrA binding site or are in an operon and coregulated with a gene that has a CtrA binding site are shown in red. Genes not regulated by CtrA are shown in white. The same thresholds were applied as in (B) to define the DivJ, PleC and DivK regulons.

manner comprised only 61, 114 and 40 genes, respectively (**S10–S12 Data**), and were thus considerably smaller than the CtrA regulon (381 genes). Surprisingly, the DivK regulon shows only little overlap with the DivJ (8%) or PleC (25%) regulons (**Fig 10B**), indicating that the roles of DivJ and PleC may go beyond the regulation of DivK activity.

While many of the genes regulated by DivJ, PleC and DivK are of unknown function, some are predicted to be involved in known, cell cycle-regulated processes. Consistent with the motility defect of the corresponding mutants (**Fig 3C**), DivJ and PleC affect the expression of genes related to flagellar assembly and rotation. Moreover, the DivJ and DivK regulons contain genes involved in c-di-GMP metabolism, and all three regulons contain various pili-related genes (**S10–S12 Data**). The major functions regulated by DivJ, PleC and DivK are thus similar to those regulated by CtrA. Nevertheless, there is only a partial overlap between the respective regulons (**Fig 10C, S9 Data and S13 Data**), with only 87% of the genes in the DivJ regulon and ~ 50% of the genes in the DivK and PleC regulons controlled and/or bound by CtrA (**Fig 10A and S10–S12 Data**). These results suggest that the regulatory effects of DivJ, PleC and DivK are not only mediated through CtrA but, directly or indirectly, also through one or more additional transcription factors. In addition, they show that the CtrA pathway as described here is a leaky pipeline, in which only part of the signal is transmitted to the next protein of the CtrA signaling network, suggesting the existence of additional regulatory components (e.g. response regulators or other transcription factors).

## CtrA is not regulated at the level of proteolysis

In *C. crescentus* and other species investigated, the activity of CtrA is tightly controlled in time and space to ensure proper cell cycle progression, based on a multi-layered regulatory network that acts at the levels of transcription, phosphorylation and protein degradation [9, 13, 63]. Having verified the transcriptional autoregulation (**Fig 9B**) and CckA-dependent

phosphorylation (**Fig 7C**) of CtrA in *H. neptunium*, we aimed to determine if this central regulator shows cell cycle-dependent changes in protein abundance. For this purpose, we first tested whether our synchronization protocol for *H. neptunium* permits the detection of fluctuations in protein levels over the course of the cell cycle, using flagellin as a marker. Previous work has shown that the flagellum of *H. neptunium* is shed at the onset of stalk formation and then re-synthesized in the nascent bud compartment [50]. Consistent with this observation, the cellular levels of flagellin decreased markedly during the swarmer-to-stalked cell transition and then increased again towards the end of the cell cycle (**Fig 11A and 11B**), confirming the validity of our analytical approach. In contrast, the levels of CtrA did not show any appreciable cell cycle-dependent changes (**Fig 11A and 11B**) under the conditions used, suggesting that it may not be subject to extensive targeted proteolysis. To further test this hypothesis, we followed the amount of CtrA in the population after inhibiting protein translation (**Fig 11E and 11F**). We indeed observed only a moderate (12±4%) decrease in CtrA levels within a period of 7 h (2.5 generation times), which was very different from the rapid degradation of $CtrA_{CC}$ in *C. crescentus* ($t_{1/2}$ = 20 min, i.e. ~0.2 generation times) (**S10 Fig**). To further investigate the apparent lack of CtrA proteolysis, we next focused on the role of RcdA and CpdR, two adapter proteins required to target CtrA to the ClpXP protease in *C. crescentus* [35–38] and other alphaproteobacteria [67, 68]. A mutant lacking CpdR, which is involved in the degradation of multiple ClpXP substrates [38], displayed mild morphological defects, including enlarged cell bodies and stalks. Inactivation of the CtrA-specific adapter RcdA, by contrast, did not have any obvious phenotypic consequences (**Fig 11C**), supporting the notion that CtrA degradation is not relevant for proper cell cycle progression in *H. neptunium*. Importantly, neither the Δ*cpdR* nor Δ*rcdA* mutant showed any increase in the average CtrA level (**Fig 11D**). Moreover, we did not observe any obvious change in the stability of CtrA in the two mutant backgrounds after the inhibition of protein synthesis (**Fig 11E and 11F**). Collectively, these results strongly suggest that unlike in *C. crescentus* [35, 36] and *S. meliloti* [67], the activity of CtrA is not regulated at the level of protein abundance in *H. neptunium*.

## Discussion

The members of most bacterial lineages divide by binary fission without prominent morphological or physiological differences between the daughter cells. Alphaproteobacteria, by contrast, have developed a variety of complex, bi- or multiphasic life cycles and thus offer the unique opportunity to study how a conserved cell cycle regulatory network has been rewired during the course of evolution to bring about fundamentally different outputs. Some of the most intricate life cycles within this order are observed in the genera *Hyphomonas* and *Hyphomicrobium*, which both divide by stalk-terminal budding. In this study, we established that the CtrA signaling cascade is essential for proper cell cycle regulation in *H. neptunium* and controls multiple central processes such as cell shape, motility, cell division and replication. Unlike in the closely related, but morphologically distinct species *C. crescentus*, the upstream part of the cascade (DivJ/PleC-DivK) has only a minor impact on CtrA activity. Moreover, there are characteristic differences in subcellular location of essential CtrA pathway components in *H. neptunium*, as CckA does not condense into polar foci but spreads throughout entire cell compartments. Finally, the activity of *H. neptunium* CtrA does not appear to be regulated at the level of protein abundance.

Interestingly, although the direct CtrA regulons of *C. crescentus* and *H. neptunium* cover similar functions, such as motility, cell division, and c-di-GMP metabolism, their precise composition differs considerably. Only slightly more than half of the genes in the direct regulon of $CtrA_{HN}$ have a homolog in *C. crescentus*, and only seven of these homologs are part of the

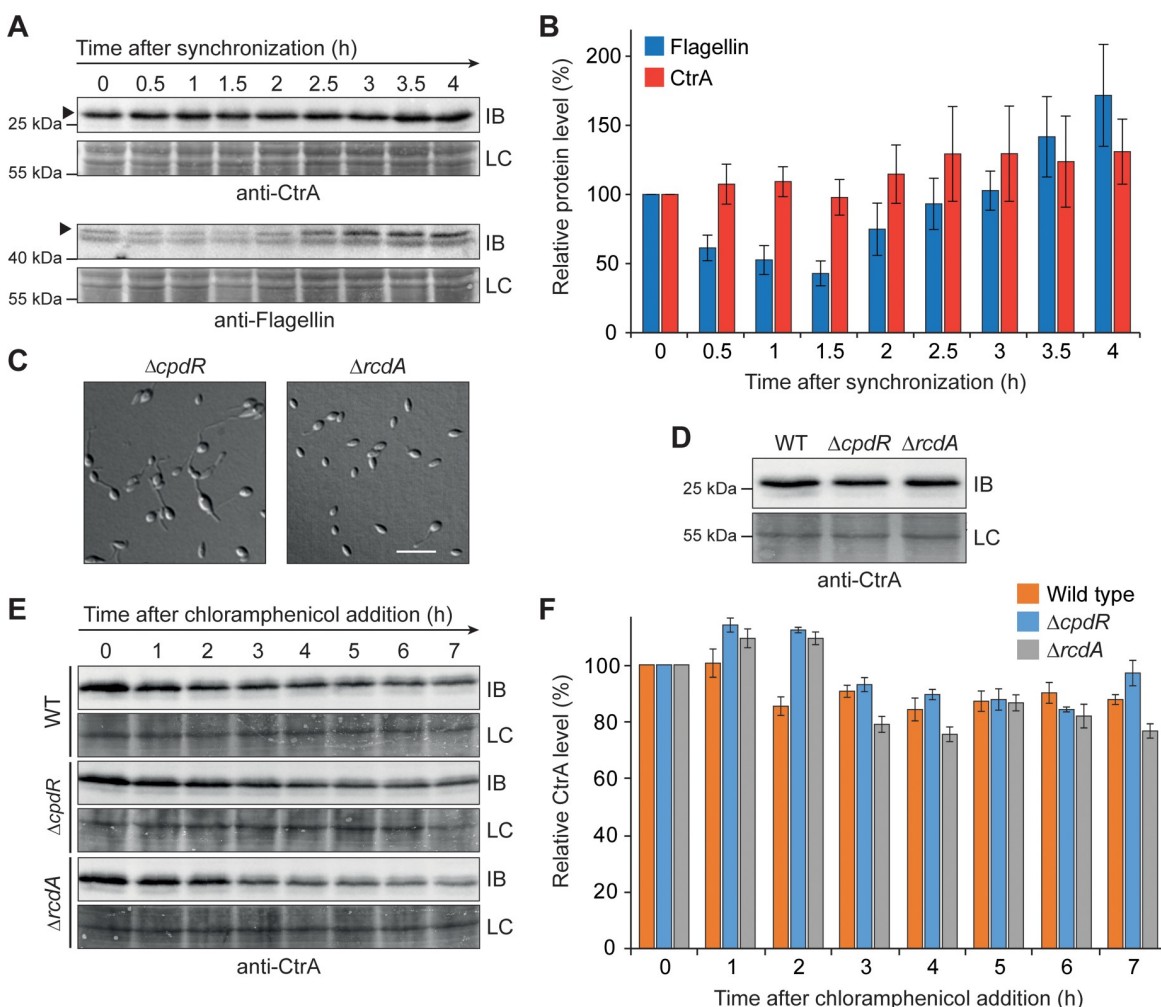

**Fig 11. CtrA levels are stable throughout the cell cycle. (A)** Immunoblot analysis showing the levels of CtrA$_{HN}$ over the course of the *H. neptunium* cell cycle. After synchronization, wild-type swarmer cells were transferred into pre-warmed medium and cultivated for 4 h. At the indicated time points, samples were taken and probed with an anti-CtrA$_{HN}$ antibody. To assess the synchrony of the cells, the same samples were additionally analyzed with an anti-flaggelin antibody. Shown are representative immunoblots (IB) and sections of the corresponding membranes stained with Amido black as loading controls (LC). The bands representing the proteins of interest are indicated by arrowheads. **(B)** Levels of CtrA and flagellin at different stages of the cell cycle. Samples were taken from a synchronized culture and analyzed by immunoblotting with anti-CtrA$_{HN}$ and anti-flagellin antibodies, as described in (A). Specific immunosignals were quantified and normalized to the total protein content of the respective lane, as detected by staining with Amido black. Values represent the mean of 7 (CtrA$_{HN}$) or 8 (flagellin) replicates. Error bars indicate the standard deviation. **(C)** DIC images of exponentially growing *H. neptunium* Δ*rcdA* (OL44) and Δ*cdpR* (OL28) cells. Scale bar: 5 μm. **(D)** Comparison of the levels of CtrA in *H. neptunium* wild type (WT), Δ*rcdA* and Δ*cdpR* cells. Cells from an exponentially growing culture were probed with anti-CtrA$_{HN}$ antibody. Shown is a representative immunoblot (IB) and a band on the corresponding membrane stained with Amido black as a loading control (LC). **(E)** Immunoblot analysis showing the stability of CtrA in the wild-type (WT), Δ*rcdA* and Δ*cdpR* backgrounds. After growth of the cells to exponential phase, translation was inhibited by addition of chloramphenicol. Samples were taken at the indicated timepoints, adjusted to the same OD$_{580}$ and probed with anti-CtrA$_{HN}$ antiserum. Shown are representative immunoblots (IB) and sections of the corresponding membranes stained with Amido black as a loading control (LC). **(F)** Levels of CtrA after translation inhibition in the wild-type (WT), Δ*rcdA* and Δ*cdpR* backgrounds. Shown are the mean intensities of the CtrA immunosignals obtained in three independent experiments of the type described in (E), corrected for cell growth after the addition of chloramphenicol. Error bars indicate the standard deviation.

direct regulon of CtrA$_{CC}$ (**S8 Data**). This striking evolutionary plasticity in the target spectrum is in line with previous results showing that CtrA, for instance, affects cell division by controlling *minCD* in *Sinorhizobium meliloti* [67] but *mipZ*, *ftsZ* and, potentially, *ftsQA* in *C. crescentus* [15]. In *H. neptunium*, by contrast, CtrA regulates the cell division genes *ftsAKQZ* but

none of the known Z-ring placement factors, such as the *H. neptunium mipZ* homolog (**S8 Data**).

The CtrA pathway also shows plasticity with respect to the importance of its components, in particular in its upstream part. In *S. meliloti*, PleC and DivK are essential, and DivJ cannot be deleted in the absence of the histidine kinase CbrA [69, 70]. In *C. crescentus*, by contrast, only DivK is essential [54], although mutants lacking DivJ or PleC show major cell polarity defects [29]. Our results now demonstrate that DivJ, PleC and DivK are all non-essential in *H. neptunium*. The observations that DivK is dispensable and its lack has no obvious phenotypic consequences is surprising. Since its interaction partner DivL is essential and critical for proper cell morphology and development, we propose the existence of one or multiple so-far unidentified factors that act in parallel to DivK to control DivL activity. The mild phenotypes of the Δ*divJ* and, even more so, the Δ*pleC* mutant raise the question of how cell polarity is ultimately determined in *H. neptunium*. A factor that may be involved is the second messenger c-di-GMP. In *C. crescentus*, the levels of c-di-GMP vary in a cell cycle-dependent manner, mostly due to changes in the activity of the hybrid guanylate cyclase/response regulator PleD, thereby critically contributing to the differential regulation of CckA activity in the two daughter cells [31, 32]. However, we found that the inactivation of PleD does not have any obvious effect in *H. neptunium*. It thus remains to be determined if c-di-GMP in fact has a role in *H. neptunium* cell cycle regulation and, if so, what protein takes over the role of PleD in coupling cell polarity to CckA stimulation. Alternatively, additional histidine kinases, similar to CbrA in *S. meliloti* [70], could be involved in marking the two opposite cell poles to control cellular asymmetry.

Interestingly, a comparison of the transcriptional profiles obtained in this study showed that the DivJ/PleC-DivK module only provides part of the signal feeding into CtrA. To obtain an integrated and quantitative view of the CtrA pathway, we devised a novel way to characterize the flow of information in signaling pathways. To gain insight into the connectivity between the different nodes, we compared all regulons described in this study and determined the number of genes that are shared between pairs of regulons as a measure of the signal that is transferred between nodes. This approach only provides a partial picture of the signal flow through the different nodes, as it does not take into account the amplitude of the transmitted signal (i.e. the difference in gene expression between regulons). Moreover, there is some uncertainty in the connectivity values, because the regulon of DivL is still unknown and the number and nature of the genes contained in the different regulons depends on the thresholds used in the analysis. The results obtained can, nonetheless, provide unprecedented insight into the CtrA regulatory pathway and predict missing factors at specific positions in the signaling cascade. For instance, this quantitative analysis verified that there must be a thus-far unknown factor next to DivK that feeds more signal into the lower part of the CtrA pathway than DivK itself (**S13 Data**). In addition, there should be a minimum of three additional factors that together provide more than half of the input into the CtrA pathway (**Fig 12**, brown ovals). These missing components may not only account for the small sizes of the DivJ, PleC and DivK regulons but also explain the low amplitudes of the transcriptional changes within these regulons. Moreover, they may be the reason why a considerable number of genes bound by CtrA were not differentially regulated upon depletion of CckA or ChpT. It will be interesting to perform similar quantitative analyses in other species and test for the completeness of the established CtrA regulatory pathways. As a first attempt, we made use of published microarray data [15, 22, 71, 72] to investigate the signal flow in the CtrA regulatory pathway of *C. crescentus* (**S14 Data**). Previous work has shown that DivK is essential in *C. crescentus* and *S. meliloti*, suggesting that the putative factor acting in parallel to DivK in *H. neptunium* is either absent or functionally less important in these species. However, in line with our findings and our preliminary quantitative analysis (**S14 Data**), there may be additional histidine kinases acting in

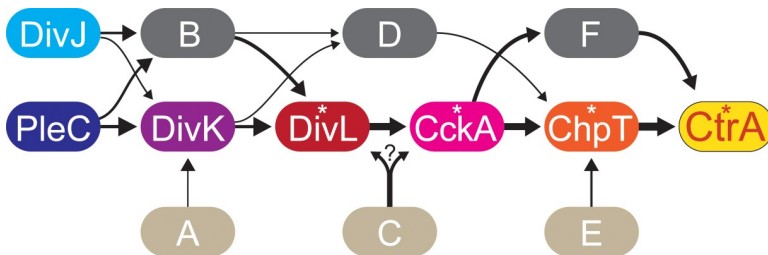

**Fig 12. Model of the cell cycle regulatory cascade controlling CtrA activity in *H. neptunium*.** DivJ, PleC and a thus-far unidentified factor (oval A; 1%) control the phosphorylation state of DivK. The role of DivK is subordinate to that of an additional regulator (oval B; ≥23%), which receives input from DivJ and PleC and potentially other proteins. DivK and the unidentified regulator transfer the majority of the signal to DivL, from where it is passed on via CckA and ChpT to CtrA. Additional unidentified factors feed into DivL and/or CckA (oval C; 44%) and ChpT (oval E; 10%), although some of the signal from DivK could also bypass DivL and CckA via an additional pathway (oval D; 2%). Finally, there may be a pathway that acts in parallel to ChpT to transfer signals from CckA to CtrA (oval F; 29%). The thickness of the lines is proportional to the fraction of the total signal reaching CtrA that is transferred via the corresponding connection (as indicated by the percentages above). Note that the sum of the input signals amounts to more than 100%, because some genes are present in the regulons of multiple input proteins. Essential proteins are marked with asterisks.

parallel to DivJ and PleC in *C. crescentus*. Indeed, previous work has shown that a *C. crescentus* mutant lacking both *pleC* and *divJ* [29] has a milder phenotype than cells depleted of DivK [54]. Moreover, DivK is still partially phosphorylated in the absence of functional DivJ and PleC [29], suggesting that $DivK_{CC}$ receives signals from at least one other, so-far unknown kinase. Further studies will be required to identify the predicted additional components of the CtrA regulatory pathway in *H. neptunium* and to determine whether these or similar factors are present in other alphaproteobacteria as well.

The results obtained in this and previous [49] work provide comprehensive insight into the spatiotemporal organization of the CtrA regulatory pathway in *H. neptunium*. Similar to the situation in *C. crescentus*, DivJ and PleC localize to opposite poles once the cell has left the swarmer stage (**Fig 13**). PleD, by contrast, differs from its *C. crescentus* homolog [73] by local-izing to the new pole of the nascent daughter cell [49], and the amount of PleD that accumu-lates at the pole, as opposed to the diffuse population in the cytoplasm, appears to be higher in *H. neptunium*. The significance of this difference in PleD localization remains unclear, as the deletion of $pleD_{HN}$ does not produce any noticeable phenotype, while $PleD_{CC}$ function is important for motility and stalk formation in *C. crescentus* [74, 75]. CckA generally localizes to similar subcellular regions in *H. neptunium* and *C. crescentus*, as it is initially distributed within the swarmer cell envelope and later condenses in the nascent swarmer cell [26, 27]. However, in *H. neptunium*, CckA does not form distinct polar foci but rather spreads through-out the respective compartments. It remains to be determined how CckA can be specifically enriched in the nascent bud without being tethered to the cell pole, especially since PodJ [12], which in *C. crescentus* helps to recruit CckA to the nascent swarmer pole via MopJ [76] and possibly DivL [25], is still polarly localized in *H. neptunium*. The involvement of PodJ in the positioning of multiple factors may explain why cells lacking PodJ have a more severe pheno-type than those lacking PleC (compare **S5 Fig** to **Fig 3**). Notably, the role of $PodJ_{HN}$ seems to go beyond that of its *C. crescentus* homolog, because a *C. crescentus* Δ*podJ* mutant has only a comparatively mild cell cycle defect [12].

Another interesting question is why *H. neptunium* cells can tolerate a more diffuse CckA localization. In *C. crescentus*, two hypotheses have been put forward to explain the condensa-tion of CckA into a polar focus. First, CckA was shown to require a high enough local concen-tration to self-interact efficiently and become active as a kinase [26, 27, 77, 78]. In addition, its

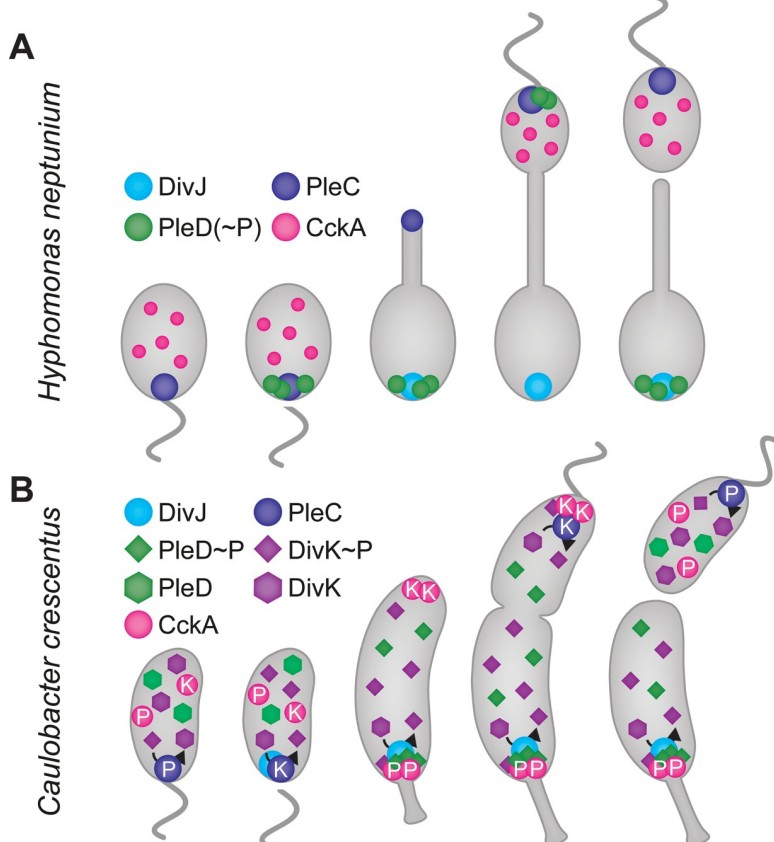

**Fig 13. Localization patterns of cell cycle regulatory proteins in *H. neptunium* and *C. crescentus*. (A)** Localization patterns in *H. neptunium*. Similar to its *C. crescentus* homolog, the *H. neptunium* histidine kinase/phosphatase PleC localizes to the flagellated pole of the swarmer cell and then relocates to the new pole once stalk formation initiates. Concomitant with the relocation of PleC, the histidine kinase DivJ localizes to the old (previously flagellated) pole, where it remains until cell division takes place. The predicted hybrid guanylate cycle/response regulator PleD first colocalizes with PleC at the flagellated pole and then relocates to the nascent bud to colocalize with DivJ. The histidine kinase CckA, by contrast, is initially dispersed throughout the swarmer cell membrane. As the cell cycle progresses, it condenses in the bud compartment but, unlike its *C. crescentus* homolog, never forms a distinct polar focus. **(B)** Localization patterns in *C. crescentus*. PleC localizes to the flagellated pole of the swarmer cell, acting as a phosphatase (P) that dephosphorylates DivK. Once the cell sheds the flagellum and starts to progress through its cell cycle, it switches from the phosphatase to the kinase (K) mode and relocates to the newly flagellated pole. DivJ, by contrast, localizes to the stalked pole, where it cooperates with PleC to maintain DivK in the phosphorylated state. CckA forms clusters that are randomly distributed throughout the swarmer cell envelope. At later stages of the cell cycle, it localizes to the stalked and flagellated poles, where it acts as a phosphatase (P) and kinase (K), respectively.

accumulation at the flagellated pole may help to establish a gradient of CtrA~P with a maximum close to the pole-proximal replication origin, thereby blocking replication initiation in the (nascent) swarmer cell [77, 79, 80]. The absence of a polar CckA focus in the bud compartment of *H. neptunium* may suggest that CckA activity is not regulated at the level of protein concentration in this species, unless its activity is triggered at considerably lower threshold levels. However, there may be other mechanisms to still establish a CtrA~P gradient in *H. neptunium*. For instance, the narrow lumen of the stalk [10] may be sufficient to slow down the diffusion of CtrA(~P) from the mother cell to the bud compartments. Another aspect that may contribute to gradient formation could be the specific and/or non-specific DNA-binding activity of CtrA(~P). A recent study in *C. crescentus* revealed that CtrA shows constrained diffusion, likely because of its interaction with the nucleoid [77]. Previous work showed that its affinity for specific DNA binding sites

increases by two orders of magnitude upon phosphorylation [81–83], and it is conceivable that phosphorylation also increases, at least to some extent, its affinity for non-specific DNA. Importantly, in *H. neptunium*, the presence of the stalk leads to an almost complete spatial separation of the two sister chromosomes at all stages of the cell cycle. Unlike in *C. crescentus*, where the two sister nucleoids form a continuum, the majority of CtrA~P molecules generated by CckA in the nascent bud may thus be trapped on the local nucleoid in the bud compartment. In contrast, dephosphorylated CtrA that is generated in the mother cell may be preferentially retained on the mother cell nucleoid, or equilibrate rapidly within the cell in case its DNA binding affinity is too low to affect its diffusional behavior. The enrichment of CtrA~P in the nascent bud may obviate the need to degrade CtrA in the mother cell to re-initiate the next cell cycle once cell division has occurred. Indeed, the observation that *H. neptunium* CtrA is not degraded in a cell cycle-dependent manner represents a striking difference between this species and other alphaproteobacteria that have been studied so far [22, 37, 67]. Its unusual morphology thus renders *H. neptunium* an excellent model system to develop and study new hypotheses on the mechanisms that underlie the control of cell differentiation by CtrA and to clarify how gradients of DNA-binding proteins such as CtrA(~P) could be established within bacterial cells.

Collectively, the comparison of *H. neptunium* and *C. crescentus* reveals significant differences in the CtrA pathway with respect to (i) the precise nature of the genes that it regulates, (ii) the importance of the individual proteins involved the pathway and, thus, the degree of redundancy at the nodes in the pathway and (iii) the localization pattern of the major polarity determinant CckA. These findings indicate a striking plasticity in the connectivity and output of the pathway among even closely related alphaproteobacterial species that likely results from their adaptation to distinct lifestyles and cell morphologies. The comprehensive and quantitative analysis of the CtrA regulatory network presented and the hypotheses developed in this work can inspire research in more well-studied model organisms and thereby contribute to a deeper understanding of the regulatory pathways mediating bacterial cell cycle regulation.

## Material and methods

### Growth conditions

The *H. neptunium* wild type LE670 (ATCC 15444) [50] and its derivatives were cultivated at 28˚C in Difco Marine Broth 2216 (BD Biosciences), supplemented with kanamycin (100/200) or triclosan (0.25/0.25) when required (μg ml$^{-1}$ in liquid/solid medium). The expression of genes placed under the control of the copper-responsive promoter $P_{Cu}$ [49] was induced with 300 μM $CuSO_4$. The *C. crescentus* wild type (NA1000) [84] and its derivatives were grown in peptone yeast extract (PYE) medium at 28˚C. When appropriate, media were supplemented with kanamycin (30/50). Gene expression from the xylose-inducible $P_{xyl}$ promoter [85] was induced with 0.3% xylose. *E. coli* strains were cultivated in LB at 37˚C, supplemented with the following antibiotics where appropriate (μg ml$^{-1}$ in liquid/solid medium): ampicillin (200/200), kanamycin (30/50), triclosan (10/20) and chloramphenicol (30/20). *E. coli* strain WM3064 was grown in the presence of 300 μM 2,6-diaminopimelic acid (DAP). To induce the expression of genes from the T7 promoter, media were supplemented with 0.5 mM isopropyl-β-D-thiogalac-topyranoside (IPTG). All strains were cultivated aerobically while shaking at 210 rpm.

### Construction of plasmids and strains

The strains, plasmids and oligonucleotides used in this study are listed in **S1–S6 Tables**. All plasmids were verified by DNA sequencing. *E. coli* TOP10 (Invitrogen) and XL1-Blue (Agilent Technologies) were used for cloning purposes. The heterologous overproduction of proteins was achieved in Rosetta 2(DE3)pLysS (Merck), whereas bacterial two-hybrid experiments

were performed in *E. coli* BTH101 (EuroMedex). *H. neptunium* cells were transformed by conjugation with *E. coli* WM3064 as described previously [49]. Genomic *in-frame* deletions in *H. neptunium* were generated by double-homologous recombination using the counter-selectable *sacB* marker [10, 86]. Briefly, *H. neptunium* cells were transformed with a non-replicating plasmid harbouring 600 bp long up- and downstream flanking regions and the first and last 18 bp of the genomic region to be deleted. Correct integration of the plasmid was confirmed by colony PCR and transformants were grown to stationary phase in non-selective media, after which they were plated (dilution 1:200) on MB agar plates supplemented with 3% sucrose. After 5–8 days of incubation at 28˚C, single colonies were re-streaked in parallel on plates containing either sucrose or the antibiotic, whose corresponding antibiotic cassette was carried on the plasmid used for construction. Individual clones growing in the presence of sucrose were subsequently tested for successful deletions by colony PCR with oligonucleotides priming outside the flanking regions used for recombination. For the construction of conditional *cckA* and *chpT* mutants, a copy of the respective gene was placed under control of $P_{Cu}$ on a non-replicating plasmid and integrated at the $P_{Cu}$ locus as previously described [49]. The native copy of the gene was then deleted as described above while expression of the ectopic copy was induced. *C. crescentus* cells were transformed by electroporation as previously described [87].

## Growth curves

Cells were grown to mid-exponential growth phase and diluted to an optical density at 580 nm ($OD_{580}$) of 0.05. Measurements were performed in a volume of 1 ml in 24-well polystyrene microtiter plates (BD Bioscience), using an Epoch2 microplate reader (BioTek). Growth was monitored over a period of 30 h at 28˚C or 30˚C with double-orbital shaking, with data points acquired at 30 min intervals. To follow growth after inhibition of translation, media were supplemented with 5 μg/ml chloramphenicol. The division time of each strain was calculated by fitting at least four replicate growth curves to a suitable growth model [88].

## Microscopy

Exponentially growing cells were immobilized on pads consisting of 1% agar and imaged with an Axio Observer.Z1 microscope (Zeiss, Germany) equipped with a Zeiss Plan-Apochromat 100x/1.46 Oil DIC objective (Zeiss, Germany). An X-Cite 120PC metal halide light source (EXFO, Canada) and ET-CFP, ET-YFP or ET-TexasRed filter cubes (Chroma, USA) were used for fluorescence detection. Images were acquired with a pco.edge sCMOS camera (PCO, Germany), recorded using VisiView (Visitron Systems, Germany) and processed using Metamorph 7.7.5 (Molecular Devices, USA) and Adobe Illustrator CS5 (Adobe Systems, USA). To stain chromosomal DNA, exponentially growing cells were mixed with 4′,6-diamidino-2-phenylindole (DAPI) at a final concentration of 4 mg/ml and then incubated for 20 min at 28˚C in the dark prior to analysis. Phenotypes were assessed by manually determining the fraction of aberrantly shaped cells in a representative population of stalked cells. Cell shape was classified as aberrant if the stalk was unusually wide or reached more than twice the normal length, if a cell had more than one stalk or if the area of the mother cell or bud exceeded the normal value by more than 1.5-fold.

## Motility assays

Exponential *H. neptunium* cultures were diluted to an $OD_{580}$ of 0.2 and dropped onto soft-agar plates consisting of 0.25% agar in 30% MB medium. The area of growth covered by each strain was measured after 6 days of incubation at 28˚C using ImageJ v1.47 [89] and compared to that obtained for the wild type and a non-motile Δ*fliL* mutant. The data shown represent

the mean of at least fourteen replicates. A one-sided Mann-Whitney U-test was performed to test for significant differences in the motility of wild-type and mutant cells. Raw data are shown in **S15 Data**.

## Flow cytometry

Cells from an exponential culture were mixed with 10 μM Vybrant® DyeCycle™ Orange Stain (Molecular Probes) and incubated for 45 min at 28˚C in the dark. Samples were analysed with a LSRFortessa analyzer (BD Biosciences). Fluorescence was excited using an Argon-green laser (514 nm) at 100 mW in combination with a 542/27 bandpass filter. Cell size was assessed based on the forward scatter, using a laser (488 nm) at 100 mW. Each strain was analyzed at least in duplicate (n = 30,000 cells per run). Cytometry data was acquired using FACSDiva™ Software 8.0 (BD Biosciences) and analyzed using FlowJo 5.4+ (FlowJo LLC, USA).

## Bacterial two-hybrid assays

The genes under study were cloned into the vectors pUT18(-C) and pK(N)T25, which carried genes encoding the T18 and T25 fragments of *Bordetella pertussis* adenylate cyclase (obtained as part of the BACTH kit from EuroMedex). After transfer of the resulting plasmids into reporter strain *E. coli* BTH101, transformants were plated on MacConkey agar (Carl Roth) containing 1% maltose. Cells from single colonies were inoculated into LB medium, grown to stationary phase, and dropped onto MacConkey agar containing 1% maltose. Interactions were indicated by a red to purple coloration of the colonies after 48 h of growth at 37˚C. A strain producing fusions of the yeast GCN4 leucin zipper regions (Zip) to the T18 and T25 fragments were included as a positive control in all assays [90]. Furthermore, for each interaction to be analyzed, cells carrying the respective pUT18(-C) derivative and empty pK(N)T25 served as a negative control.

## Protein purification

To purify $His_6$-$DivJ_{HN}$, $His_6$-$PleC_{HN}$, $His_6$-$DivK_{HN}$, $His_6$-$CckA_{HN}$-RR, $His_6$-$ChpT_{HN}$ and $His_6$-$CtrA_{HN}$, *E. coli* Rosetta 2(DE3)pLysS was transformed with plasmids pOL135 ($DivJ_{HN}$), pOL208 ($PleC_{HN}$), pOL134 ($DivK_{HN}$), pJR75 ($CckA_{HN}$-RR), pOL207 ($ChpT_{HN}$) or pOL145 ($CtrA_{HN}$), respectively, and grown at 37˚C (or at 18˚C in the case of $PleC_{HN}$) in LB medium supplemented with 0.5% glucose. When the culture had reached an $OD_{580}$ of 0.5–0.7, expression of the respective protein was induced with 0.5 mM IPTG. Once the culture had reached an $OD_{580}$ of 2.5, the cells were harvested by centrifugation for 10 min at 4,500 g, washed twice in buffer A (50 mM $Na_2HPO_4$, 300 mM NaCl, 10 mM imidazole, adjusted to pH 8.0 with HCl) and stored at -80˚C until further use. To purify the proteins, the cells were thawed on ice, resuspended in buffer A containing 10 μg/ml DNase I and 100 μg/ml PMSF, and disrupted by two to four passages through a French press at 16,000 psi. After centrifugation of the lysate at 30,000 g for 30–60 min at 4˚C to remove cell debris, the supernatant was applied to a 5 ml HisTrap HP column (GE Healthcare) equilibrated with buffer A. The column was washed with 5 column volumes (CV) of buffer A, and protein was eluted with a linear imidazole gradient obtained by mixing buffer A with buffer B (50 mM $Na_2HPO_4$, 300 mM NaCl, 500 mM imidazole, adjusted to pH 8.0 with HCl) at a flow rate of 1 ml/min using an ÄKTA purifier 10 FPLC system (GE Healthcare). After analysis by SDS-PAGE, elution fractions containing the protein of interest were pooled and subjected to a second purification step (except in the case of $His_6$-CckA-$RR_{HN}$). To this end, the pooled fractions were loaded onto a 120 ml Superdex 200 16/60 (GE-Healthcare, USA) size exclusion column equilibrated with buffer C (50 mM $Na_2HPO_4$, 300 mM NaCl, adjusted to pH 8.0 with HCl), and protein was eluted with 1 CV of

buffer C at a flow rate of 0.5 ml/min. Fractions containing the desired protein at high concentrations and purity (as visualized by SDS-PAGE) were pooled and dialyzed against buffer D (50 mM $Na_2HPO_4$, 300 mM NaCl, 5% glycerol, adjusted to pH 8.0 with HCl). The protein solution was then aliquotted, snap-frozen in liquid nitrogen and stored at -80˚C until further use.

His$_6$-CckA$_{CC}$ and His$_6$-CckA-KD$_{CC}$ were purified as described previously [91]. *E. coli* Rosetta 2(DE3)pLysS containing plasmid pMSB3 (CckA$_{CC}$) or pMSB2 (CckA-KD$_{CC}$) was treated as described above with the following changes. After harvest, the cells were resuspended in lysis buffer (20 mM Tris-HCl, 0.5 M NaCl, 10% glycerol, 20 mM imidazole, 0.1% Triton X-100, pH 7.9) containing 10 µg/ml DNase I and 100 µg/ml PMSF. The HisTrap HP column (GE Healthcare) was equilibrated with wash buffer (20 mM HEPES-KOH, 0.5 M NaCl, 10% glycerol, 20 mM imidazole, 0.1% Triton X-100, pH 8.0), and protein was eluted with a linear gradient obtained by mixing wash buffer and elution buffer (20 mM HEPES-KOH, 0.5 M NaCl, 10% glycerol, 250 mM imidazole, pH 8.0) at a flow rate of 2 ml/min. Fractions containing the desired proteins at high concentrations and purity were pooled, dialyzed against storage buffer (10 mM HEPES-KOH, 50 mM KCl, 10% glycerol, 0.1 mM EDTA, 1 mM dithiothreitol, pH 8.0), snap-frozen and stored at -80˚C until further use.

To obtain HNE_0264 (Flagellin) for antibody generation, a His$_6$-SUMO-tagged derivative of the protein was overproduced in *E. coli* Rosetta 2(DE3)pLysS transformed with plasmid pJR74 and purified as described for His$_6$-CckA$_{CC}$ and His$_6$-CckA-KD$_{CC}$ with the following changes. The lysis buffer contained 0.3 M KCl instead of 0.5 M NaCl, and protein was eluted with 10 CV of a linear gradient of 30–250 mM imidazole at a flow rate of 1 ml/min. After the first purification step on a 5 ml HisTrap HP column, the His$_6$-SUMO-tag was cleaved off in buffer containing 1 mM DTT by treatment with His$_6$-Ulp1 protease for 2 h at 4˚C. The protein solution was then applied again to a 5 ml HisTrap HP column as described above. Untagged HNE_0264 was recovered from the flow through, dialyzed against storage buffer (50 mM Tris-HCl, 150 mM KCl, 10% glycerol, pH 8.0), snap-frozen and stored at -80˚C until further use.

## Phosphotransfer profiling

Phosphotransfer was analyzed essentially as described previously [91]. To promote the autophosphorylation of His$_6$-CckA$_{CC}$ and His$_6$-CckA$_{CC}$ -KD, the proteins were incubated at 30˚C in storage buffer (10 mM HEPES-KOH, 50 mM KCl, 10% glycerol, 0.1 mM EDTA, pH 8.0) containing 2 mM DTT, 5 mM MgCl$_2$, 500 µM ATP and 5 µCi [γ$^{32}$P]-ATP (∼3,000 Ci/mmol, Hartmann Analytic). The autophosphorylation of DivJ and PleC was achieved by incubation of the proteins at 28˚C in buffer R1 (50 mM Na$_2$HPO$_4$, 300 mM NaCl, pH 8.0) or buffer R2 (50 mM Tris/HCl, 50 mM NaCl, pH 7.6) containing 5 mM MgCl$_2$ and 5 µCi [γ$^{32}$P]-ATP. The phosphotransfer reactions were started by mixing phosphorylated His$_6$-PleC$_{HN}$, His$_6$-CckA$_{HN}$, His$_6$-CckA$_{HN}$-RR (autophosphorylated for 45 min) or His$_6$-DivJ$_{HN}$ (autophosphorylated for 30 min) with the acceptor protein(s) to be tested, which were diluted to a concentration of 3–5 µM in the respective storage buffer supplemented with 5 mM MgCl$_2$. The mixtures were incubated for 5 min at 30˚C (for reactions including CckA, ChpT and/or CtrA) or for 90 sec at 28˚C (for reactions including DivJ, PleC and/or DivK). After addition of the same volume of 2x sample buffer (500 mM Tris/HCl pH 6.8, 8% SDS, 40% glycerol, 400 mM β-mercaptoethanol, 0.01% bromophenol blue), the proteins were loaded, without boiling, onto an 8–16% TGX Precast Protein Gel (Bio-Rad) and separated for 50 min at a constant voltage of 150V at room temperature. After removal of unincorporated ATP in the dye front with a scalpel, the gel was transferred to a plastic bag and exposed to a phosphor screen overnight. The phosphor screen was scanned with a Storm 840 or 860 system (Molecular Dynamics) at a resolution of 50 dots/cm. Each phosphotransfer assay was repeated three times.

## Chromatin immunoprecipitation coupled to deep sequencig (ChIP-seq)

A culture of exponentially growing *H. neptunium* wild type cells at an $OD_{580}$ of 0.5 was treated with paraformaldehyde (1% final concentration) in 10 mM sodium phosphate buffer (pH 7.6) at RT for 10 min to achieve crosslinking. Subsequently, the cultures were incubated for an additional 30 min on ice and washed three times in phosphate-buffered saline (PBS, pH 7.4). The resulting cell pellets were frozen in liquid nitrogen and stored at -80˚C. After resuspension in TES buffer (10 mM Tris-HCl pH 7.5, 1 mM EDTA, 100 mM NaCl) containing 10 mM of DTT, the cells were incubated in the presence of Ready-Lyse lysozyme solution (Epicentre, Madison, WI) for 10 minutes at 37˚C according to the manufacturer's instructions. The lysates were sonicated (Bioruptor Pico) at 4˚C using 15 bursts of 30 sec to shear DNA fragments to an average length of 0.3–0.5 kbp and cleared by centrifugation at 14,000 rpm for 2 min at 4˚C. After adjustment of the volumes (relative to the protein concentration) to 1 ml using ChIP buffer (0.01% SDS, 1.1% Triton X-84 100, 1.2 mM EDTA, 16.7 mM Tris-HCl pH 8.1, 167 mM NaCl,) containing protease inhibitors (Roche), the lysates were pre-cleared with 80 µl of protein-A agarose (Roche) and 100 µg BSA. 2% of each pre-cleared lysate was kept as total input (negative control) samples. The rest of the pre-cleared lysates was incubated overnight at 4˚C with polyclonal antibodies targeting $CtrA_{HN}$ or $GcrA_{CC}$ [21] (1:1,000 dilution). The immunocomplexes were captured after incubation with Protein-A agarose (pre-saturated with BSA) during a 2 h incubation at 4˚C and subsequently washed with low-salt washing buffer (0.1% SDS, 1% Triton X-100, 2 mM EDTA, 20 mM Tris-HCl pH 8.1, 150 mM NaCl), with high-salt washing buffer (0.1% SDS, 1% Triton X-100, 2 mM EDTA, 20 mM Tris-HCl pH 8.1, 500 mM NaCl), with LiCl washing buffer (0.25 M LiCl, 1% NP-40, 1% deoxycholate, 1 mM EDTA, 10 mM Tris-HCl pH 8.1) and finally twice with TE buffer (10 mM Tris-HCl pH 8.1, 1 mM EDTA). Immunocomplexes were eluted from the Protein-A agarose beads with two times 250 µl elution buffer (SDS 1%, 0.1 M NaHCO₃, freshly prepared) and then, in parallel to the total input samples, incubated overnight with 300 mM NaCl at 65˚C to reverse the crosslinks. The samples were then treated with 2 µg of Proteinase K for 2 h at 45˚C in 40 mM EDTA and 40 mM Tris-HCl (pH 6.5). DNA was extracted using phenol:chloroform:isoamyl alcohol (25:24:1), ethanol-precipitated using 20 µg of glycogen as a carrier and resuspended in 50 µl of DNAse/RNAse-free water.

Immunoprecipitated chromatin was used to prepare sample libraries for deep-sequencing at Fasteris SA (Geneva, Switzerland). ChIP-seq libraries were prepared using the DNA Sample Prep Kit (Illumina) according to the manufacturer's instructions. Single-end runs (50 cycles) were performed on an Illumina Genome Analyzer IIx or HiSeq2000, yielding several million reads. The single-end sequence reads (stored as FastQ files) were mapped against the genome of *Hyphomonas neptunium* ATCC 15444 (CP000158) using Bowtie version 0.12.9 (-qS best parameters, http://bowtie-bio.sourceforge.net/). ChIP-seq read sequencing and alignment statistics are summarized in **S2 Data**. After conversion of the output into standard genomic position format files (BAM format; using Samtools; http://samtools.sourceforge.net), the data were imported into SeqMonk version 0.34.1 (Babraham Bioinformatics; http://www.bioinformatics.babraham.ac.uk/projects/seqmonk/) to build ChIP-seq normalized sequence read profiles. Briefly, the genome was subdivided into 50 bp probes, and for every probe, we calculated the number of reads per probe as a function of the total number of reads (per million, using the Read Count Quantitation option). Analyzed data as shown in **Figs 7 and 8** are provided in **S3 Data** (50 bp resolution) and **S4 Data** (10 bp resolution). Using the web-based analysis platform Galaxy (https://usegalaxy.org), CtrA and GcrA ChIP-seq peaks were called with MACS2 [92] relative to the total input DNA samples. The q-value (false discovery rate, FDR) cut-off for called peaks was 0.05. Peaks were rank-ordered according to fold-enrichment (**S5 Data**), and

peaks with a fold-enrichment values >2 were retained for further analysis. Consensus sequences common to the top 80 enriched CtrA$_{HN}$-associated loci were identified by scanning peak sequences (+/- 75 bp relative to the coordinates of the peak summit) for conserved motifs using MEME (http://meme-suite.org/) [93]. Binding sites of CtrA close to the origin of replication were identified using the previously identified origin region [66]. To locate CtrA binding sites upstream of transcriptional units in the motility islands, operons were predicted with DOOR$^2$ [94].

## Expression profiling of mutants by RNA-Seq and analysis of regulons

To assess the global expression changes in the *ΔdivJ*, *ΔpleC* and *ΔdivK* mutants and the conditional *cckA* and *chpT* mutants as compared to wild-type cells, two biological replicates of each strain (including the wild type) were grown to mid-exponential growth phase. After dilution of each culture to an OD$_{580}$ of 0.15–0.2, the cells were washed two times in MB medium, harvested by centrifugation, snap-frozen in liquid nitrogen and stored at -80˚C. For the conditional mutants and a wild-type replicate, two aliquots of cells were harvested: one that was grown in the presence of CuSO$_4$ for 24 h, and one that was depleted of the respective protein by growth in the absence of CuSO$_4$ for 24 h. RNA extraction, cDNA library preparation, sequencing as well as the normalization of the expression data were performed by Vertis Biotechnologie AG (Germany). In order to identify genes that were differentially expressed in the *ΔdivJ*, *ΔpleC* or *ΔdivK* deletion mutants or the *cckA* or *chpT* conditional mutants, the log$_2$-fold changes of the normalized expression values (RPKM) were compared between the wild-type and mutant strains. For the *cckA* and *chpT* conditional mutants, genes were considered as differentially expressed when they showed at least a 1.3 log$_2$-fold change in their expression with a p-value of at least 0.05 in a paired T-test. For the *ΔdivJ*, *ΔpleC* and *ΔdivK* mutants, a 0.5 log$_2$-fold change and a p-value of at least 0.25 were used as thresholds. Genes represented by only a few transcripts (RPKM values below 25) were excluded from further analyses. The evaluation as well as the statistical analysis of the expression data was performed in Excel 2016 (Microsoft). For visual representation of the relevant expression data, genes were grouped with Cluster 3.0 [95] using the City-block distance option and visualized in Java TreeView 1.1.6r4 [96].

Genes that were differentially expressed in the *cckA* and/or *chpT* conditional mutants and featured a CtrA binding site in their immediate upstream region, in the gene itself or in a gene in the same operon, as predicted by DOOR$^2$ [94], were defined as the CtrA direct regulon. To quantify the information flow in the CtrA pathway, the overlap between the different regulons was determined. When a gene was differentially regulated such that it passed the log$_2$-fold thresholds described above, it was counted as contributing to the information flow. In case of the *cckA* and *chpT* conditional mutants, genes located in an operon with a regulated gene were also counted as part of the regulons. As the *ΔdivJ*, *ΔpleC* and *ΔdivK* mutants showed only very moderate transcriptional changes, we also considered genes in the information flux analysis that were left out of the regulons of DivJ, PleC and DivK because of a too high p-value. The 89 genes constituting the CtrA direct regulon were normalized to 100% and from the number of overlapping genes, the percentage of the signal leading up to CtrA was calculated at every arrow. Missing histidine kinases were postulated based on a difference between the overlap between the respective regulon and the CtrA direct regulon and that of the regulon of the preceding node, as well as from evaluating the signal input and output in total and at every node. These two approaches gave highly similar results for all missing histidine kinases.

## Synchronization of *H. neptunium*

Cells were synchronized by filtration as described previously [97]. In brief, *H. neptunium* was grown to late exponential phase (OD$_{580}$ of 0.6), pelleted (3,000 g at 4˚C), resupended in ice-

cold PBS and filtered through 1.2 μm pore-sized nitrocellulose filters (Merck-Millipore). The flow-through was collected on ice, filtered again (pore size 0.8 μm, Merck-Millipore). The flow-through of the second filtration step, which was highly enriched in swarmer cells, was centrifuged and the resulting pellet was resuspended in pre-warmed media to an $OD_{580}$ of 0.4. The culture was then incubated at 28˚C and samples were withdrawn at the indicated intervals. The synchrony of the culture was assessed using microscopy.

## Antibody generation and immunoblot analysis

Polyclonal antibodies against $His_6$-CtrA and flagellin (purified as described above) were raised in rabbits by Eurogentec (Belgium) and Davids Antibodies (Germany), respectively. Immuno-blot analysis was performed on cultures that had been harvested by centrifugation and resus-pended in 2x SDS sample buffer such that each sample was normalized to the $OD_{580}$ of the culture. Cells were lyzed by incubation at 95˚C for 10 min and proteins were separated on 11–15% SDS-polyacrylamide gels and transferred to a polyvinylidene fluoride (PVDF) membrane (Millipore, Germany). Immunodetection was performed with polyclonal rabbit sera targeting $CtrA_{HN}$, $CtrA_{CC}$ [13], GFP (Sigma Aldrich) (all 1:10,000) and flagellin (1:2,000) according to standard procedures, using goat-anti-rabbit conjugated to horseradish peroxidase (PerkinEl-mer, USA) as a secondary antibody. Immunocomplexes were visualized using the Western Lightning plus-ECL chemiluminescent reagent (PerkinElmer, USA) in a ChemiDoc MP imag-ing system (Bio-Rad, USA). ImageLab 5.0 (Bio-Rad, USA) and Adobe Illustrator CS5 (Adobe Systems, USA) were used for acquisition, quantification and further image processing.

To visualize fluctuations in the levels of CtrA and flagellin over the course of the cell cycle, samples were taken from the same culture every 30 min from the onset of synchrony. Samples were loaded on separate gels for CtrA and flagellin immunodetection. Both experiments were repeated 7 or 8 times in total, with samples stemming from the same synchronization event. The amounts of CtrA and flagellin were normalized for $t_0$ and all bands were normalized for the total amount of protein present in the entire lane (as measured by Amido black staining) in order to enable a comparison between the different timepoints. Raw data are shown in **S15 Data**.

## Determination of CtrA stability *in vivo*

In order to probe the stability of CtrA *in vivo*, *H. neptunium* or *C. crescentus* cells were grown to mid-exponential growth phase and protein synthesis was blocked by addition of 5–10 μg/ml chloramphenicol for *H. neptunium* [49] and 100 μg/ml chloramphenicol for *C. crescentus*. At the indicated intervals, 1 or 2 ml samples were withdrawn for immunoblot analysis and loaded on a gel after normalization to the $OD_{580}$ of the sample (as described above). CtrA was detected using the α-CtrA antibody (1:10000) raised in this study. Signals were quantified with the ImageLab 5.0 software (Bio-Rad, USA). For each strain, immunoblots of two clones from three independent experiments were analysed. As the focus was on the total amount of CtrA present in the culture, growth curves recorded in the presence of 5 μg/ml chloramphenicol (as described above) were used to correct for the dilution of CtrA due to cell growth after inhibi-tion of translation. Raw data are shown in **S15 Data**.

## Bioinformatics

All DNA sequences were retrieved from KEGG [98] and protein sequences were obtained from KEGG or UniProt [99]. The SMART database [100] was used to identify (transmem-brane) domains in histidine kinases and response regulators. Homologs of proteins from *C. crescentus* were identified via BLAST [101]. The annotation of genes in the different regulons

was performed manually, using the annotations in UniProt. COG categories were obtained via the Microbial Genomic context Viewer [102] and COG function classes using the table provided on the NCBI COG website (ftp://ftp.ncbi.nih.gov/pub/wolf/COGs/COG0303/cogs.csv).

## Supporting information

**S1 Fig. Cross-complementation shows the (partial) conservation of function of cell cycle-relevant genes between *C. crescentus* and *H. neptunium*. (A)** DIC images of *C. crescentus* mutants with in-frame deletions in $divJ_{CC}$ (YB3202) or $pleC_{CC}$ (UJ506) or temperature-sensitive mutations in $divK_{CC}$ (LS3570), $divL_{CC}$ (KR635) or $cckA_{CC}$ (CckATS1) expressing $divJ_{HN}$ (OL133), $pleC_{HN}$ (OL135), $divK_{HN}$ (OL137), $divL_{HN}$ (MvT81) or $cckA_{HN}$ (OL179), respectively, from a xylose-inducible promoter (under restrictive conditions). *C. crescentus* wild-type (CB15N) cells are shown for comparison. **(B)** Quantification of the fraction of stalked cells in cultures of a $\Delta pleC_{CC}$ mutant carrying a xylose-inducible copy of $pleC_{HN}$ (OL135) in the absence or presence of inducer. **(C)** *C. crescentus* strain carrying a temperature-sensitive variant of $CtrA_{CC}$ (LS2195) at the restrictive (37˚C) and permissive (28˚C) temperature. **(D)** Dominant negative effect of $CtrA_{HN}$ expressed from a xylose-inducible promoter ($P_{xyl}$) in cells producing either a temperature-sensitive (OL128; at the permissive temperature) or the wild-type (OL130) form of $CtrA_{CC}$. **(E)** Immunoblots showing the degradation of $CtrA_{CC}$ upon induction of $CtrA_{HN}$ in *C. crescentus*. *C. crescentus* strains carrying a temperature-sensitive (OL128) or wild-type (OL130) allele of $ctrA_{CC}$ and an ectopic copy of $ctrA_{HN}$ under the control of a xylose-inducible promoter were grown in the absence (-) or presence (+) of xylose and analyzed with anti-$CtrA_{HN}$ and anti-$CtrA_{CC}$ antibodies. Cells of the *C. crescentus* $ctrA^{ts}$ (LS2195) and wild-type ($WT_{CC}$; CB15N) strains and the *H. neptunium* wild-type strain ($WT_{HN}$; LE670) were analyzed as controls. All cultures analyzed in panels A-E were grown at 28˚C, unless stated otherwise. Cells were withdrawn from exponential cultures after depletion and/or induction of the respective proteins for 24 h. Scale bars: 5 μm.
(TIF)

**S2 Fig. Expression of $divJ_{HN}$ can complement the phenotype of an *H. neptunium* $\Delta divJ_{HN}$ mutant .** An *H. neptunium* $\Delta divJ_{HN}$ mutant carrying an ectopic copy of $divJ_{HN}$ under the control of a copper-inducible promoter (OL123) was grown for 24 h in copper-containing medium and subjected to DIC microscopy. The percentage of stalked cells in the culture and the division time are shown on the right. Scale bar: 5 μm.
(TIF)

**S3 Fig. *H. neptunium* cells still segregate chromosomal DNA after depletion of DivL, CckA or ChpT.** *H. neptunium* strains carrying conditional alleles of $divL_{HN}$ (OL177), $cckA_{HN}$ (OL161) or $chpT_{HN}$ (OL152) were grown for 24 h in the absence of inducer. Chromosomal DNA was stained with DAPI prior to imaging. Wild-type cells are shown for comparison. Scale bar: 5 μm. The percentage of cell bodies that show a DAPI signal is given in the bottom right corner of each fluorescence image.
(TIF)

**S4 Fig. Polar localization of DivJ and PleC depends on SpmX and PodJ respectively.** DivJ-Venus does not condense into distinct foci in cells lacking SpmX (OL36), whereas it shows the typical polar localization in the wild-type background (OL146). Similarly, PleC-eYFP foci are observed only sporadically in cells lacking PodJ (OL166), whereas they form normally in the wild-type background (OL151). Scale bars: 5 μm.
(TIF)

**S5 Fig. Lack of *podJ*, but not of *spmX*, leads to morphological defects.** Shown are DIC images of *H. neptunium* Δ*spmX* (OL34) and Δ*podJ* (OL35) cells. A quantification of the proportion of stalked cells with aberrant morphologies is given below the images. Scale bar: 5 μm.
(TIF)

**S6 Fig. CckA-Venus supports normal growth and is stably expressed. (A)** Growth of an *H. neptunium* strain expressing *cckA-venus* in place of the native *cckA* gene (OL2). The growth of wild-type (LE760) cells is shown for comparison. Data represent the average of five independent experiments. **(B)** Immunoblot showing the accumulation of CckA-Venus. Samples of the strains analyzed in (A) were probed with anti-GFP antibodies. The full-length CckA-Venus fusion is indicated by an orange arrowhead. Cleaved Venus is indicated by a black arrowhead.
(TIF)

**S7 Fig. CckA-KD$_{CC}$ can phosphorylate CtrA$_{HN}$ directly when CckA-RR$_{HN}$ is absent.** CckA-KD$_{CC}$ was autophosphorylated for 45 min at 30˚C. Subsequently, the indicated proteins (marked with pluses) were combined and incubated for 5 min at 30˚C. After termination of the reactions by addition of SDS sample buffer, proteins were separated by SDS-PAGE and radioactivity was detected by phosphor imaging.
(TIF)

**S8 Fig. The CtrA level decreases upon depletion of CckA and ChpT. (A)** Immunoblot showing the levels of CtrA after depletion of CckA or ChpT. Conditional *H. neptunium* mutants carrying copper-inducible copies of *cckA* (OL161) or *chpT* (OL152) were cultivated for 24 h in the absence of inducer and probed with anti-CtrA$_{HN}$ antibodies. Wild-type cells were analyzed for comparison. A representative section of the membrane stained with Amido black is shown as a loading control. **(B)** Quantification of the levels of CtrA after depletion of CckA or ChpT. The conditional *cckA* and *chpT* mutants analyzed in (A) were grown for 24 h in the presence (+ Cu) and absence (- Cu) of inducer and subjected to immunoblot analysis with anti-CtrA$_{HN}$ antibodies. The signals were quantified and normalized to the signal obtained for wild-type control cells. Data represent the average of three biological replicates, each of which was analyzed in triplicate. Error bars indicate the standard deviation.
(TIF)

**S9 Fig. CtrA regulation predominantly affects genes of unknown function as well as genes involved in cellular processes and signaling. (A)** Overview of the proportion of different COG categories among the 381 genes that are differentially expressed upon depletion of CckA and ChpT (see **Fig 8A**). Only genes with an RPKM value of >25, a p-value of <0.05 and a log$_2$-fold change in expression of >2 were taken in account. **(B)** Overview of the proportion of different COG categories among the 285 genes contained in the transcriptional units that are immediately adjacent to the 222 CtrA binding sites identified in this study (see **Fig 8B**). **(C)** Venn diagram showing the number of genes (single or in an operon) bound by CtrA (in blue) and the number of genes present in the entire CtrA regulon (in red). The intersection of these two gene sets defines the direct CtrA regulon and comprises 94 genes. **(C)** Overview of the abundance of different COG categories among the 94 genes contained in the direct CtrA regulon (see **Fig 8D**).
(TIF)

**S10 Fig. The half-life of CtrACC in *C. crescentus* is considerably shorter than the generation time.** (A) Immunoblot showing decrease in the level of CtrA$_{CC}$ in *C. crescentus* after the inhibition of translation. Chloramphenicol was added to an exponentially growing culture of the *C. crescentus* wild-type (CB15N) strain. Samples were taken at the indicated time points

and probed with anti-CtrA$_{CC}$ antibodies. Shown are a representative immunoblot (IB) and a section of the membrane stained with Amido black as a loading control (LC). To calculate the half-life of CtrA$_{CC}$ in *C. crescentus*, the signals were quantified and fitted to a single-exponential function. Data represent the mean of the values obtained from two independent immunoblots (± SD).
(TIF)

**S1 Table. *E. coli* strains used in this study.**
(PDF)

**S2 Table. *H. neptunium* strains used in this study.**
(PDF)

**S3 Table. *C. crescentus* strains used in this study.**
(PDF)

**S4 Table. General plasmids used in this study.**
(PDF)

**S5 Table. Plasmids generated in this study.**
(PDF)

**S6 Table. Oligonucleotides in this study.**
(PDF)

**S1 Data. Total CtrA regulon, comprising all genes that are differentially expressed upon depletion of CckA and/or ChpT.** All genes whose expression levels differed at least 2.5-fold (1.3 log$_2$-fold) from the values obtained for the wild type were classified as differentially expressed. Only genes with a p-value <0.05 (paired T-test) and an RPKM value of >25 were taken into account. Genes that are highlighted in yellow are part of the direct regulon. Genes are subdivided into functional categories based on manual annotation using the UniProt database.
(XLSX)

**S2 Data. Summary of the ChIP-seq sequencing and alignment statistics.** Shown are the number of reads and mapped reads and the fold genome coverage obtained in the CtrA and GcrA Chip-seq experiments and the respective total input (negative control) samples.
(XLSX)

**S3 Data. Distribution of the CtrA and GcrA ChIP-seq reads over the *H. neptunium* chromsome (50 bp resolution).** The reads obtained for the CtrA and GcrA ChIP-seq samples and the respective total input (negative control) samples were mapped onto the *H. neptunium* genome sequence. Shown are the normalized read frequencies in RPM (reads per million reads) determined for a series of 50 bp windows covering the entire length of the chromosome. The corresponding HNE locus tags, feature strands and feature descriptions are indicated.
(XLSX)

**S4 Data. Distribution of CtrA and GcrA ChIP-seq reads within specific regions of the *H. neptunium* chromosome (10 bp resolution).** Shown are the normalized read frequencies (in RPM, reads per million reads) obtained for the CtrA and GcrA ChIP-seq experiments and the respective total input (negative control) in a series of consecutive 10 bp windows covering the replication origin region, the *ctrA* promotor region, and the motility island of the *H. neptunium* chromosome. The corresponding HNE locus tags, feature strands and feature descriptions are also reported.
(XLSX)

**S5 Data. MACS2 peak detection analysis of the CtrA and GcrA ChIP-seq experiments.** CtrA ("CtrA Chip-seq") and GcrA ("GcrA ChIP-seq") ChIP-seq peaks were called using MACS2 software. The same analysis was performed on the respective total input DNA samples as a negative control. The q-value (false discovery rate, FDR) cut-off for called peaks was 0.05. Peaks are rank-ordered according to fold-enrichment and highlighted in red if the reads constituting them are significantly (at least twofold) enriched in the ChIP-seq sample. The peak intervals, peak summits, closest genes and statistics (-$\log_{10}$ of the p-values and q-values as well as the fold enrichment) are indicated for all detected peaks. (XLSX)

**S6 Data. CtrA binding sites on the *H. neptunium* binding sites as determined by ChIP-seq analysis.** The start, end, length and summit of the peaks in the distribution of the sequences pulled down by the CtrA antibody are given for each observed CtrA binding site. The fold enrichment indicates how much the sequences constituting each peak were enriched in the ChIP-seq analysis with the CtrA antibody compared to the total input sample. The locus tag and annotation of genes potentially affected by CtrA binding are given for each peak. (XLSX)

**S7 Data. Direct CtrA regulon.** The direct CtrA regulon is composed of genes that are differentially expressed upon CckA and/or ChpT depletion (**S1 Data**) and have a CtrA binding site as observed by ChIP-seq (**S6 Data**) or of genes that are in an operon with such genes. Genes that are present in the same operon (predicted by DOOR$^2$) are highlighted in the same color. The functional classification of these genes was performed by manual annotation using the Uniprot database and is also reflected by the COG categories. (XLSX)

**S8 Data. Comparison between the direct CtrA regulons of *C. crescentus* and *H. neptunium*.** The genes and functions present in the direct CtrA regulons of *H. neptunium* and *C. crescentus* (Laub et al., 2002) were compared. Genes whose closest homolog is contained in the CtrA regulons of both species are highlighted in green. (XLSX)

**S9 Data. Comparison of the genes in the direct CtrA regulon in different backgrounds.** A comparison of the expression levels of the genes in the direct CtrA regulon shows that the $\log_2$-fold change in gene expression is more pronounced in the strains depleted of CckA or ChpT than in strains lacking *divJ*, *pleC* or *divK*. Genes that are present in the same operon (predicted via DOOR$^2$) are highlighted in the same color. (XLSX)

**S10 Data. DivJ regulon.** The DivJ regulon is composed of genes that are differentially expressed (threshold $\log_2$-fold change: 0.5) upon deletion of *divJ*. Only genes with a p-value <0.25 (paired T-test) and an RPKM value of >25 were taken into account. For each gene, it is indicated if this gene is bound by CtrA in the ChIPseq experiment (CtrA ChIP) and if it is present in the direct CtrA regulon. Genes that are present in the same operon (predicted by DOOR$^2$) are highlighted in the same color. Genes are subdivided into functional categories based on manual annotation using the UniProt database. (XLSX)

**S11 Data. PleC regulon.** The PleC regulon is composed of genes that are differentially expressed (threshold $\log_2$-fold change: 0.5) upon deletion of *pleC*. Only genes with a p-value <0.25 (paired T-test) and an RPKM value of >25 were taken into account. For each gene, it is

indicated if this gene is bound by CtrA in the ChIPseq experiment (CtrA ChIP) and if it is present in the CtrA direct regulon. Genes that are present in the same operon (predicted by DOOR[2]) are highlighted in the same color. Genes are subdivided into functional categories based on manual annotation using the UniProt database.
(XLSX)

**S12 Data. DivK regulon.** The DivK regulon is composed of genes that are differentially expressed (threshold $\log_2$-fold change: 0.5) upon deletion of *divK*. Only genes with a p-value <0.25 (paired T-test) and an RPKM value of >25 were taken into account. For each gene, it is indicated if this gene is bound by CtrA in the ChIPseq experiment (CtrA ChIP) and if it is present in the CtrA direct regulon. Genes that are present in the same operon (predicted by DOOR[2]) are highlighted in the same color. Genes are subdivided into functional categories based on manual annotation using the UniProt database.
(XLSX)

**S13 Data. Quantification of overlap between the direct CtrA, ChpT, CckA, DivK, PleC and DivJ regulons enables a reconstruction of the signal flow through the CtrA pathway.** Shown is an overview of the genes from the direct CtrA regulon that are present in the ChpT, CckA, DivK PleC and DivJ regulons. The thresholds for inclusion in a regulon were a $\log_2$-fold change of 1.3 for CckA and ChpT and $\log_2$-fold change of 0.5 for DivJ, PleC and DivK. Based on the overlap between the different regulons, the signal input and output at the different nodes and the signal flux between the different nodes in the regulatory cascade were quantified as a percentage of the total signal input into CtrA.
(XLSX)

**S14 Data. Analysis of the signal flow in the CtrA regulatory pathway of *C. crescentus*.** The table shows the overlap between the CtrA regulon and the PleC, DivJ, CckA and ChpT regulons in *C. crescentus*.
(XLSX)

**S15 Data. Raw data.** The file provides the raw data for the motility assays and Western blot analyses.
(XLSX)

## Acknowledgments

We thank Julia Rosum, Stephanie Stede and Maritha Lippmann for excellent technical assistance, Timur Sander for the initial characterization of strains, and Marlène Birk for the construction of plasmids. Juliane Kühn and Susan Schlimpert are acknowledged for advice. We are grateful to Silvia González Sierra for help with the FACS analysis, Lotte Søgaard-Andersen and Michael Bölker for support, and Dorota Skotnicka for advice on the work with radioactivity.

## Author Contributions

**Conceptualization:** Oliver Leicht, Martin Thanbichler.

**Data curation:** Oliver Leicht, Muriel C. F. van Teeseling, Gaël Panis.

**Formal analysis:** Oliver Leicht, Muriel C. F. van Teeseling, Gaël Panis.

**Funding acquisition:** Patrick H. Viollier, Martin Thanbichler.

**Investigation:** Oliver Leicht, Muriel C. F. van Teeseling, Gaël Panis, Celine Reif, Heiko Wendt.

**Methodology:** Gaël Panis.

**Project administration:** Patrick H. Viollier, Martin Thanbichler.

**Supervision:** Patrick H. Viollier, Martin Thanbichler.

**Validation:** Muriel C. F. van Teeseling.

**Visualization:** Oliver Leicht, Muriel C. F. van Teeseling, Gaël Panis, Celine Reif, Martin Thanbichler.

**Writing – original draft:** Muriel C. F. van Teeseling, Martin Thanbichler.

**Writing – review & editing:** Oliver Leicht, Muriel C. F. van Teeseling, Gaël Panis, Celine Reif, Heiko Wendt, Patrick H. Viollier, Martin Thanbichler.

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
