## [Decision Letter · Decision Letter 0]

27 Aug 2019

Dear Martin,

Thank you very much for submitting your Research Article entitled 'Integrative and quantitative view of the CtrA regulatory network in a stalked budding bacterium' to PLOS Genetics. Your manuscript was fully evaluated at the editorial level and by independent peer reviewers. The reviewers appreciated the attention to an important problem, but raised some substantial concerns about the current manuscript. Based on the reviews, we will not be able to accept this version of the manuscript, but we would be willing to review again a much-revised version. We cannot, of course, promise publication at that time.

If you decide to revise the manuscript for further consideration at PLOS Genetics, please aim to resubmit within the next 60 days, unless it will take extra time to address the concerns of the reviewers, in which case we would appreciate an expected resubmission date by email to plosgenetics@plos.org.

[LINK]

We are sorry that we cannot be more positive about your manuscript at this stage. Please do not hesitate to contact us if you have any concerns or questions.

Yours sincerely,

Josep Casadesús

Section Editor: Prokaryotic Genetics

PLOS Genetics

Reviewer's Responses to Questions

**Comments to the Authors:**

Reviewer #1: Sequencing of many bacterial genomes will undoubtly generate questions about the extend of extrapolations we can make for functions in diverse bacteria, based on homology with model species. Therefore the study reported here by the Thanbichler team is really important for the field of molecular microbiology. The topic is very interesting and the conclusions are well supported by the numerous data included in this manuscript. It is an impressive work of high quality and the manuscript is well written. I have several comments and requests for a revised version of the manuscript.

The abstract is too vague and too general, and should be written with more precision, for example reporting that divK is not essential, that CckA localization is completely different from the one in C. crescentus and that CtrA half life is much longer...

Line 114 (Table 1) : I would include the E-value of the next (i.e. second) closest homolog of each C. crescentus protein in H. netunium genome, to really show that the identified H. neptunium is probably an ortholog. Also, are they best reciprocal matches?

Line 123 (Figure 3B and 3C) : I suggest to add the statistical analysis of the data to compare the swarm area of the different strains

Line 146 (Figure S3) : it is not obvious that DNA is segregated to the nascent daughter cell for the ∆cckA PCu-cckA uninduced condition. Is it also true for only one bacterium in the ∆chpT PCu-chpT uninduced strain? It would be nice to show more bacteria or to have a quantification

Lines from 155 (Figure 5A,B,C,D) : what is the proportion of bacteria with a polar fusion to DivJ, SpmX, PleC or PodJ? I think that this data should be incorporated.

Line 166 (Figure 5E,F) : does CckA contain one or several predicted transmembrane segments like in C. crescentus?

Line 184 : I think that reference to Figure S1A is wrong here

Line 187 : Figure S7 mentions that ChpT is dispensable if CckARR is absent. I think that this information should be removed because, in the current state of the manuscript, it does not bring a new interesting data

Line 195 : Figure 8A,H : I would remove the WT line (not necessary)

Line 197 : the authors report genes up- or down-regulated in the absence of CckA and/or ChpT as "the global CtrA regulon". I think that this is not correct, because CckA and ChpT could control other (unknown) regulators.

Line 201 : 286 genes are mentionned here while it is 284 genes in Figure 8F. In Figure 8F I would indicate the numbers 381 and 284(or 286) outside the Venn ensemble, while keeping "94" in the intersection between these ensembles.

Line 214 : in Figure 8B,E,G would it possible to place the COGs in the same order? indeed, several COGs are shown in black, and it is thus difficult to know which fraction corresponds to which COG.

Line 221 : reference to Figure 8G should be Figure 8H?

Line 232 : add a reference to Jung et al., in press, as in line 598, for the identification of the origin of replication

Lines 244-249 : I completely disagree with the choice made to include differences with p-values <0.25 and log2-fold difference of >0.5. Moreover, this is not crucial to deliver the main message behind this analysis. I propose to remove this part of the analysis

Line 261 : the data reported here clearly show that the the upstream and downstream parts of the DivK-CtrA network are clearly uncoupled for a large fraction of target genes, but it does not shown that "only part of the signal is transmitted to the next protein at EACH step" since CckA and ChpT-controlled regulons are very correlated

Line 263 : I would not argue that CtrA is not controlled at the level of proteolysis because it could be controlled in other conditions, not tested here. I suggest to mention that the half-life of CtrA in C. crescentus and H. neptunium are very different, at least along the cell cycle in the conditions tested here.

Line 289 : Figure 11E,F are the data in Fugure 11F really extracted from the blots shown in Fig. 11E? In Figure 11E it seems that CtrA level is more strongly decreased in the ∆rcdA strain compared to WT and ∆cpdR, but maybe isn't it reproducible?

Lines 289-290 : why not adding "as in C. crescentus" after "CtrA is not regulated"?

From line 344 : I would remove the Discussion about the flow of signal and its quantification. This is the only really weak part of the manuscript and it does not contribute to the main conclusion of the paper. Such considerations should also be removed from the abstract. I would also remove Figure 12.

Lines 629-634 : I would remove this part about quantification of "signal" (see above).

Reviewer #2: Hyphomonas neptunium is an emerging model to study the diversity of pathways regulating the bacterial cell cycle and the Thanbichler lab pioneers to reach this new goal. In this study, they made an extremely comprehensive analysis of the impact of the CtrA response regulator and of its post-transcriptional regulators (kinases, proteases…) on the regulation of the quite original cell cycle of this bacterium (it divides through buds that form at the tip of a stalk-like structure) and made fruitful comparisons with the well-known Caulobacter model. They constructed a long series of deletion or depletion mutants for each of the regulators upstream of CtrA and perfomed RNA-Seq analysis, which were combined with ChIP-Seq experiments using CtrA antibodies, to define the direct CtrA regulon in this bacterium. In addition, they engineered strains encoding fluorescently-labeled regulators to look at their subcellular localization as a function of the cell cycle and used BTH and in vitro phosphorylation assays, to reveals how and where key regulators are phosphorylated and active in cells. Altogether, their results demonstrate that the network of genes regulated by CtrA diverged significantly between the two compared bacteria, despite the fact that they are close relatives at the genome level. They also show that the regulatory network controlling CtrA activity and CtrA levels mostly includes the same actors as in Caulobacter, while their function has, again, significantly evolved, illustrating extraordinary plasticity. Their model finally suggests the existence of novel regulators that still need to be identified in this emerging model, providing interesting perspectives. The only things that are regrettable, is that they could not obtain a CtrA-depletion strain due to technical constrains (that are fully understandable) and that experiments addressing the impact of CtrA on replication control (through binding to the chromsosomal origin) in H. neptunium remain limited in this study (see “Major comment” below). Overall, the manuscript is however very informative and carefully written.

Major comments:

1) Lines 145-147: Authors could test if mutants over-replicate their chromosome much more directly through an estimation of the ori/ter ratio using Q-PCR. This would be informative, since one of the major functions of CtrA, besides its activity as a transcription factor, is to control replication initiation.

2) I note that many promoters that are supposedly bound to CtrA in vivo (from ChIP-Seq data in Fig.8F) are not significantly mis-regulated in cckA and chpT mutants (from RNA-Seq = genes described as “Entire regulon” in Fig.8F). Could this mean that certain genes (up to 284 of them) are regulated by CtrA independently of the phosphorylation status of CtrA? Did they still find many promoters with a CtrA binding motif in this set of genes? It may be something interesting to discuss.

Minor comments:

1) Author Summary (lines 18-19): Maybe you should also mention that CtrA is not only a transcriptional regulator, but also a repressor of replication initiation that binds to the chromosomal origin.

2) Although Fig.1A is essentially a review of the CtrA regulatory network in Caulobacter, I think it is important to keep it in, considering the complexity of this network.

3) Line 43: replace “offspring” by “offsprings”

4) Lines 172-174: can you please add a few references there

5) Line 270: change “course the cell cycle” for “course of the cell cycle”

6) Line 272: delete “at”

7) Line 317: change for “The observations that DivK is dispensible and that its lack has no obvious…”

8) Line 404: “C. crescentus” (typo)

9) Line 507: delete “a”

10) Line 1400: add “.” in front of “The”

Reviewer #3: This manuscript addresses systems-level connections among an important and intensely studied group of regulatory proteins in H Neptunium, a species that serves as a comparator with the related and far better understood model species C. crescentus. It is an unusually broad and far-reaching study, and assembles a large amount of experimental data around a model that provides some interesting insight into a complex signaling pathway. A cynic would say that the main conclusion in comparing these two related species is that they are mostly the same but a bit different, and that this is not a very interesting result. However, the manuscript strives to go steps beyond this and describe how the details of pathway connectivity can change while its overall structure is maintained. Understanding such evolutionary turning points are important, particularly in light of other studies in more distantly related bacteria, which show that the CtrA pathway is no longer essential and reduced to a more limited role in the cell cycle. Clearly the pathway has undergone a substantial degree of evolutionary change, but we don’t understand how this is possible for a core pathway that is essential for life in some species.

The manuscript is mostly successful in striving to describe the wiring of this similar-but-different system, but it is missing two key points. One is positive evidence that the phosphorylation state of H. Neptunium CtrA has an effect on cell division / cell physiology, and the other is better points of comparison with C. crescentus vis a vis the leaky pipeline model. These points are described in greater detail in the major comments section below.

MAJOR COMMENTS

Figure S8, Figure 11, Figure 12: The manuscript shows that H. Neptunium CtrA is an essential gene and provides further evidence supporting the idea that it acts as a master regulator of gene expression and chromosome replication, as it does in Caulobacter. The manuscript also shows that H Neptunium CtrA is not degraded by CpdR/RcdA mediated proteolysis, and that CtrA protein levels are stable over the cell cycle. Having ruled out proteolysis as a regulatory mechanism, the other likely mode of regulation for CtrA is phosphorylation. This type of mechanism is very strongly implied by the author’s phosphotransfer signaling model in Figure 12, the in vitro phosphotranfer experiments in earlier figures, and in many other places throughout the manuscript text. But a major gap in the data is that CtrA phosphorylation states are never explored experimentally. Several potential ways of solidifying this important part of the manuscript’s conclusions are suggested as options below:

+ In Figure S8, it would be helpful to determine whether CtrA is differentially phosporylated relative to protein level under CckA and ChpT depletion conditions. Residual phosphorylation might also provide some evidence for parts of the “leaky pathway” model in Figure 12.

+ In Figure 11E, it might be informative to determine whether the half-life of phospho-CtrA in H. Neptunium is less than the half-life of the protein. This would suggest that the phosphorylation level of the protein is changed during its lifetime, strongly implying regulation by phosphosignaling.

+ Determining the fate of Phospho-CtrA in synchronized cultures and comparing that signal to CtrA protein levels (as shown in Figure 11A) may help to demonstrate phospho-regulation.

+ It might be possible to see phenotypes with the expression of a mutant form of CtrA, for example the phosphomimetic D51E and de-phospho D51A variants. Perhaps one of these will have a strong phenotype (cell shape, chromosome number, etc…) relative to wildtype CtrA, which would provide some evidence for the phosphoregulation model.

Figure 8F: Is it unusual to find that only 1/3 of the genes that have a CtrA binding site nearby are part of the “direct CtrA regulon” under CckA and ChpT? Is this further evidence for the leaky pipeline model? It would be helpful to know how this compares with the Caulobacter CtrA regulon (as defined by the genes under direct control of CckA / ChpT) and the array of CtrA binding sites in the Caulobacter genome.

L260: The meaning of “part of the signal is transmitted to the next protein at each step” isn’t clear and could have multiple interpretations. Do you mean that these proteins have multiple downstream effectors? Or are there multiple upstream signals? This statement should be restated more clearly. As with the previous comment, this interpretation would be strengthened by including a comparison with Caulobacter (if the data is present in the literature). What percentage of genes in Caulobacter DivJ / PleC / DivK regulons are shared with the Caulobacter CtrA regulon? More broadly, does the leaky pipeline model presented in this manuscript claim that the pathway is more leaky in H. Neptunium than in Caulobacter? If so, what is the evidence for this?

The abstract says that the “essential core pathway is reduced”, which probably refers to the finding that DivK is essential in C. crescentus but non-essential in H Neptunium. But the model in Figure 12 includes another protein, “Box B”, which may work in parallel with DivK, and may itself be essential or synthetic lethal with DivK. If either of these scenarios for Box B turn out to be true, it does not seem correct to say that the essential core pathway is reduced compared to Caulobacter.

Figure 12: The methods and interpretations related to this figure are interesting. Are there any other publications that have used this kind of data to generate something that looks like a pathway flux analysis or is this a novel method?

Figure 3C, S5: the use of the term “aberrant” is vague. In the methods section, provide more explanation of what qualifies as aberrant. How long does a stalk need to be in order to be considered aberrant? How does the calculation work for cell body shape?

MINOR COMMENTS

Figure 8 B,E, G: The pie chart diagrams and most of the information they contain are not particularly helpful. I suggest moving them to supplementary data.

L219-221: The manuscript shows a correlation between the presence of CtrA biding sites and changes in expression for these particular genes, but the connection has not been directly demonstrated. This is suggestive, but does not amount to proof. Please adjust the tone of this statement accordingly. Also, it seems the data these lines are referring is shown more effectively in 8H, not 8G.

L30: Please change “this right timing” to “the correct timing”.

L36: It is a bit confusing to think of it as “just a kinase and response regulator”, since the system is tied to input signals and output responses. Maybe “…or two components systems (TCSs) when the context is limited to signaling kinases and downstream response regulators.”

Figure S9: Please change “considerable” in title to “considerably”.

Figure 11D: Please attach “WB” and “LC” labels to the images, as done in 11E.

L235: “does not only provide insight…” Please re-state more clearly.

Figure 12: Text sections in figure legends and main text refer to rectangles or boxes, but the shapes are more like ovals. Perhaps “component A,B, C…” is better?

Figure 8F: Why is it 284 genes instead if the 286 genes identified in 8E?

Reference #74 is from BioRxiv. Is this permissible?

The meaning of the asterisk in Figure 12 is not explained in the figure legend.

File S9: CckA towards ChpT is 57% in field F60 and 60% in C32.

**Have all data underlying the figures and results presented in the manuscript been provided?**

Reviewer #1: Yes

Reviewer #2: Yes

Reviewer #3: Yes

PLOS authors have the option to publish the peer review history of their article (what does this mean?). If published, this will include your full peer review and any attached files.

Reviewer #1: No

Reviewer #2: No

Reviewer #3: No

---

## [Decision Letter · Decision Letter 1]

10 Mar 2020

Dear Martin,

Thank you very much for submitting a revised version of your Research Article entitled 'Integrative and quantitative view of the CtrA regulatory network in a stalked budding bacterium' to PLOS Genetics. Your revised manuscript was evaluated by the same reviewers who had reviewed the first version of the manuscript, and one of them has identified some aspects that should be improved. I therefore ask you to modify the manuscript according to the reviewer's recommendations.

I hope to receive your revised manuscript within the next 30 days. If you anticipate any delay in its return, we would ask you to let us know the expected resubmission date by email to plosgenetics@plos.org.

[LINK]

Yours sincerely,

Josep Casadesús

Section Editor: Prokaryotic Genetics

PLOS Genetics

Reviewer's Responses to Questions

**Comments to the Authors:**

Reviewer #3: Abstract: I still have problems with “essential core of the pathway is reduced….” statement in the abstract, for the same reasons that I stated in my first round of comments. I think it is better to avoid “reduced” and use other language. The essential elements of the pathway differ?

The stated reasoning for why 2/3 of genes in CtrA regulon might not vary in a CckA/ChpT depletion is tenable, but weak. I can imagine tight-binding of CtrA-P outlasting depletion or see-sawing effects in different cell types explaining the observed result for some of the genes with CtrA binding sites, but not 2/3 of them. The idea that there is differential binding of CtrA versus Phospho CtrA is plausible but I’m not sure if its supported by evidence from the literature. Generally speaking, doesn’t this result provide further support for a braided signaling network like the one shown in Figure 12?

Unfortunately we still don’t have proof that CtrA is phosphorylated in vivo or if those phosphorylation levels are relevant to H. neptunium physiology, and this is a weakness of the manuscript. This discussion point is also relevant in relation to reviewer #2’s note about the possibility that many (or indeed most?) of the genes in the CtrA regulon may be regulated by CtrA independent of its phosphorylation state. Overall, however, the in vitro phosphotransfer data and consistency in genetic phenotypes indicate that phospohtransfer through this pathway in vivo is very likely, and this is enough to support the manuscript’s final models.

I agree with the authors’ feeling that Figure 12 is important, even though it is a frustratingly complicated part of the manuscript and it includes incomplete and high p-value type information. For me, the interconnected “braids” of genetic circuitry in a core cell cycle pathway is a major take-home message of the work, and it is important for understanding the large non-overlap of some of the Venn diagrams presented earlier in the work. The effort to relate this analysis to Caulobacter literature is appreciated.

**Have all data underlying the figures and results presented in the manuscript been provided?**

Reviewer #3: Yes

PLOS authors have the option to publish the peer review history of their article (what does this mean?). If published, this will include your full peer review and any attached files.

Reviewer #3: Yes: Grant R Bowman

---

## [Editor Report · Decision Letter 2]

19 Mar 2020

Dear Martin,

I am pleased to inform you that your manuscript entitled "Integrative and quantitative view of the CtrA regulatory network in a stalked budding bacterium" has been editorially accepted for publication in PLOS Genetics. Congratulations!

Best regards! Take care!

Josep

Josep Casadesús

Section Editor: Prokaryotic Genetics

PLOS Genetics

Comments from the reviewers (if applicable):

**Data Deposition**

http://datadryad.org/submit?journalID=pgenetics&manu=PGENETICS-D-19-01258R2

**Press Queries**

---

## [Editor Report · Acceptance letter]

16 Apr 2020

PGENETICS-D-19-01258R2 

Integrative and quantitative view of the CtrA regulatory network in a stalked budding bacterium 

Dear Dr Thanbichler, 

We are pleased to inform you that your manuscript entitled "Integrative and quantitative view of the CtrA regulatory network in a stalked budding bacterium" has been formally accepted for publication in PLOS Genetics! Your manuscript is now with our production department and you will be notified of the publication date in due course.

With kind regards,

Kaitlin Butler

PLOS Genetics

On behalf of:
